# Bridging the Imitation Gap
# by Adaptive Insubordination

**Luca Weihs**[*,1], **Unnat Jain**[*,2,†], **Iou-Jen Liu**[2], **Jordi Salvador**[1],
**Svetlana Lazebnik**[2], **Aniruddha Kembhavi**[1], **Alexander Schwing**[2]
[1]Allen Institute for AI,  [2]University of Illinois at Urbana-Champaign
{lucaw, jordis, anik}@allenai.org
{uj2, iliu3, slazebni, aschwing}@illinois.edu

## Abstract

In practice, imitation learning is preferred over pure reinforcement learning whenever it is possible to design a teaching agent to provide expert supervision. However, we show that when the teaching agent makes decisions with access to privileged information that is unavailable to the student, this information is marginalized during imitation learning, resulting in an "imitation gap" and, potentially, poor results. Prior work bridges this gap via a progression from imitation learning to reinforcement learning. While often successful, gradual progression fails for tasks that require frequent switches between exploration and memorization. To better address these tasks and alleviate the imitation gap we propose 'Adaptive Insubordination' (ADVISOR). ADVISOR dynamically weights imitation and reward-based reinforcement learning losses during training, enabling on-the-fly switching between imitation and exploration. On a suite of challenging tasks set within gridworlds, multi-agent particle environments, and high-fidelity 3D simulators, we show that on-the-fly switching with ADVISOR outperforms pure imitation, pure reinforcement learning, as well as their sequential and parallel combinations.

## 1 Introduction

Imitation learning (IL) can be remarkably successful in settings where reinforcement learning (RL) struggles. For instance, IL has been shown to succeed in complex tasks with sparse rewards [8, 47, 44], and when the observations are high-dimensional, *e.g.*, in visual 3D environments [31, 54]. To succeed, IL provides the agent with consistent expert supervision at every timestep, making it less reliant on the agent randomly attaining success. To obtain this expert supervision, it is often convenient to use "privileged information," *i.e.*, information that is unavailable to the student at inference time. This privileged information takes many forms in practice. For instance, in navigational tasks, experts are frequently designed using shortest path algorithms which access the environment's connectivity graph [*e.g.*, 19]. Other forms of privilege include semantic maps [*e.g.*, 60, 13], the ability to see into "the future" via rollouts [61], and ground-truth world layouts [7]. The following example shows how this type of privileged information can result in IL dramatically failing.

**Example 1** (Poisoned Doors). Suppose an agent is presented with $N \geq 3$ doors $d_1, \ldots, d_N$. As illustrated in Fig. 1 (for $N = 4$), opening $d_1$ requires entering an unknown fixed code of length $M$. Successful code entry results in a reward of $1$, otherwise the reward is $0$. Since the code is unknown to the agent, it would need to learn the code by trial and error. All other doors can be opened without a code. For some randomly chosen $2 \leq j \leq N$ (sampled each episode), the reward behind $d_j$ is $2$ but for all $i \in \{2, \ldots, N\} \setminus \{j\}$ the reward behind $d_i$ is $-2$. Without knowing $j$, the optimal policy is to always enter the correct code to open $d_1$ obtaining an expected reward of $1$. In contrast, if the expert

---

[*]denotes equal contribution by LW and UJ; [†]work done, in part, as an intern at Allen Institute for AI

35th Conference on Neural Information Processing Systems (NeurIPS 2021).

is given the privileged knowledge of the door $d_j$ with reward 2, it will always choose to open this door immediately. It is easy to see that an agent without knowledge of $j$ attempting to imitate such an expert will learn to open a door among $d_2, \ldots, d_N$ uniformly at random obtaining an expected return of $-2 \cdot (N-3)/(N-1)$. In this setting, training with reward-based RL after a 'warm start' with IL is strictly worse than starting without it: the agent needs to unlearn its policy and then, by chance, stumble into entering the correct code for door $d_1$, a practical impossibility when $M$ is large.

To characterize this imitation failure, we show that training a student to imitate a teacher who uses privileged information results in the student learning a policy which marginalizes out this privileged information. This can result in a sub-optimal, even uniformly random, student policy over a large collection of states. We call the discrepancy between the teacher's and student's policy the *imitation gap*. To bridge the imitation gap, we introduce **Ad**aptive **In**subor**d**ination (ADVISOR). ADVISOR adaptively weights imitation and RL losses. Specifically, throughout training we use an auxiliary actor which judges whether the current observation is better treated using an IL or a RL loss. For this, the auxiliary actor attempts to reproduce the teacher's action using the observations of the student at every step. Intuitively, the weight corresponding to the IL loss is large when the auxiliary actor can reproduce the teacher's action with high confidence.

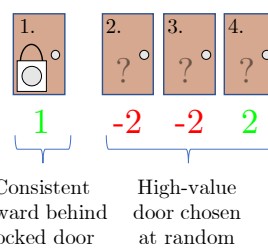

Figure 1: POISONEDDOORS.

We study the benefits of ADVISOR on thirteen tasks, including 'POISONEDDOORS' from Ex. 1, a 2D "lighthouse" gridworld, a suite of tasks set within the MINIGRID environment [8, 9], Cooperative Navigation with limited range (COOPNAV) in the multi-agent particle environment (MPE) [43, 38], and two navigational tasks set in 3D, high visual fidelity, simulators of real-world living environments (POINTNAV in AIHABITAT [54] and OBJECTNAV in ROBOTHOR [31, 14]). Our results show that,
- the imitation gap's size directly impacts agent performance when using modern learning methods,
- ADVISOR is *performant* (outperforming IL and RL baselines), *robust*, and *sample efficient*,
- ADVISOR can succeed even when expert supervision is partially corrupted, and
- ADVISOR can be easily integrated in existing pipelines spanning diverse observations (grids and pixels), actions spaces (discrete and continuous), and algorithms (PPO and MADDPG).

## 2  Related Work

A series of methods [*e.g.*, 41, 65, 3, 55] have made off-policy deep Q-learning stable for complex environments like Atari Games. Several high-performance (on-policy) policy-gradient methods for deep-RL have also been proposed [56, 42, 34, 68, 61]. For instance, Trust Region Policy Optimization (TRPO) [56] improves sample-efficiency by safely integrating larger gradient steps. Proximal Policy Optimization (PPO) [58] employs a clipped variant of TRPO's surrogate objective and is widely adopted in the deep RL community. We use PPO as a baseline in our experiments.

As environments get more complex, navigating the search space with only deep RL and simple heuristic exploration (such as $\epsilon$-greedy) is increasingly difficult. Therefore, methods that imitate expert (*i.e.*, teacher) supervision were introduced. A popular approach to imitation learning (IL) is Behaviour Cloning (BC), *i.e.*, use of a supervised classification loss between the policy of the student and expert agents [53, 2]. However, BC suffers from compounding errors. Namely, a single mistake of the student may lead to settings that have never been observed in training [51]. To address this, Data Aggregation (DAgger) [52] trains a sequence of student policies by querying the expert at states beyond those that would be reached by following only expert actions. IL is further enhanced by, *e.g.*, hierarchies [33], improving over the expert [5, 4, 27], bypassing any intermediate reward function inference [24], and/or learning from experts that differ from the student [18, 26, 16]. Importantly, a sequential combination of IL and RL, *i.e.*, pre-training a model on expert data before letting the agent interact with the environment, performs remarkably well. This strategy has been applied in a wide range of applications – the game of Go [61], robotic and motor skills [49, 30, 48, 50], navigation in visually realistic environments [19, 12], and web & language based tasks [21, 11, 59, 67].

More recent methods mix expert demonstrations with the agent's own rollouts instead of using a sequential combination of IL followed by RL. Chemali and Lazaric [6] perform policy iteration from expert and on-policy demonstrations. DQfD [23] initializes the replay buffer with expert episodes

and adds rollouts of (a pretrained) agent. They weight experiences based on the previous temporal difference errors [55] and use a supervised loss to learn from the expert. For continuous action spaces, DDPGfD [66] analogously incorporates IL into DDPG [35]. POfD [28] improves by adding a demonstration-guided exploration term, *i.e.*, the Jensen-Shannon divergence between the expert's and the learner's policy (estimated using occupancy measures). THOR uses suboptimal experts to reshape rewards and then searches over a finite planning horizon [62]. Zhu et al. [72] show that a combination of GAIL [24] and RL can be highly effective for difficult manipulation tasks.

Critically, the above methods have, implicitly or explicitly, been designed under certain assumptions (*e.g.*, the agent operates in an MDP) which imply the expert and student observe the same state. Different from the above methods, we investigate the difference of privilege between the expert policy and the learned policy. Contrary to a sequential, static, or rule-based combination of supervised loss or divergence, we train an auxiliary actor to adaptively weight IL and RL losses. To the best of our knowledge, this hasn't been studied before. In concurrent work, Warrington et al. [69] address the imitation gap by jointly training their teacher and student to adapt the teacher to the student. For our applications of interest, this work is not applicable as our expert teachers are fixed.

Our approach attempts to reduce the imitation gap directly, assuming the information available to the learning agent is fixed. An indirect approach to reduce this gap is to enrich the information available to the agent or to improve the agent's memory of past experience. Several works have considered this direction in the context of autonomous driving [10, 20] and continuous control [17]. We expect that these methods can be beneficially combined with the method that we discuss next.

## 3 ADVISOR

We first introduce notation to define the imitation gap and illustrate how it arises due to 'policy averaging.' Using an 'auxiliary policy' construct, we then propose ADVISOR to bridge this gap. Finally, we show how to estimate the auxiliary policy in practice using deep networks. In what follows we will use the terms teacher and expert interchangeably. Our use of "teacher" is meant to emphasize that these policies are (1) designed for providing supervision for a student and (2) need not be optimal among all policies.

### 3.1 Imitation gap

We want an agent to complete task $\mathcal{T}$ in environment $\mathcal{E}$. The environment has states $s \in \mathcal{S}$ and the agent executes an action $a \in \mathcal{A}$ at every discrete timestep $t \geq 0$. For simplicity and w.l.o.g. assume both $\mathcal{A}$ and $\mathcal{S}$ are finite. For example, let $\mathcal{E}$ be a 1D-gridworld in which the agent is tasked with navigating to a location by executing actions to move left or right, as shown in Fig. 2a. Here and below we assume states $s \in \mathcal{S}$ encapsulate historical information so that $s$ includes the full trajectory of the agent up to time $t \geq 0$. The objective is to find a policy $\pi$, *i.e.*, a mapping from states to distributions over actions, which maximizes an evaluation criterion. Often this policy search is restricted to a set of feasible policies $\Pi^{\text{feas.}}$, for instance $\Pi^{\text{feas.}}$ may be the set $\{\pi(\cdot; \theta) : \theta \in \mathbb{R}^D\}$ where $\pi(\cdot; \theta)$ is a deep neural network with $D$-dimensional parameters $\theta$. In classical (deep) RL [41, 42], the evaluation criterion is usually the expected $\gamma$-discounted future return.

We focus on the setting of partially-observed Markov decision processes (POMDPs) where an agent makes decisions without access to the full state information. We model this restricted access by defining a *filtration function* $f : \mathcal{S} \to \mathcal{O}_f$ and limiting the space of feasible policies to those policies $\Pi_f^{\text{feas.}}$ for which the value of $\pi(s)$ depends on $s$ only through $f(s)$, *i.e.*, so that $f(s) = f(s')$ implies $\pi(s) = \pi(s')$. We call any $\pi$ satisfying this condition an $f$-*restricted policy* and the set of feasible $f$-restricted policies $\Pi_f^{\text{feas.}}$. In a gridworld example, $f$ might restrict $s$ to only include information local to the agent's current position as shown in Figs. 2c, 2d. If a $f$-restricted policy is optimal among all other $f$-restricted policies, we say it is $f$-*optimal*. We call $o \in \mathcal{O}_f$ a *partial-observation* and for any $f$-restricted policy $\pi_f$ we write $\pi_f(o)$ to mean $\pi_f(s)$ if $f(s) = o$. It is frequently the case that, during training, we have access to a teacher policy which is able to successfully complete the task $\mathcal{T}$. This teacher policy may have access to the whole environment state and thus may be optimal among all policies. Alternatively, the teacher policy may, like the student, only make decisions given partial information (*e.g.*, a human who sees exactly the same inputs as the student). For flexibility we will define the teacher policy as $\pi_{f^{\text{teach}}}^{\text{teach}}$, denoting it is an $f^{\text{teach}}$-restricted policy for some filtration function

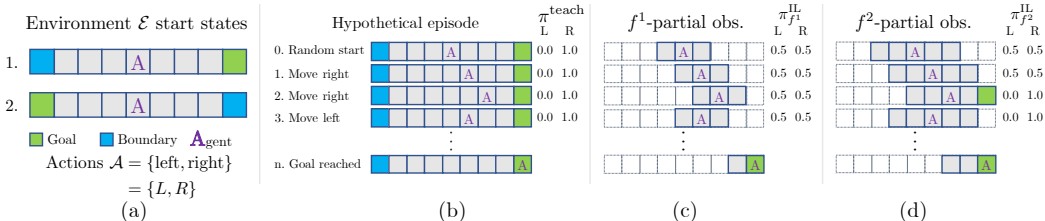

Figure 2: **Effect of partial observability in a 1-dimensional gridworld environment.** (a) The two start states and actions space for 1D-Lighthouse with $N = 4$. (b) A trajectory of the agent following a hypothetical random policy. At every trajectory step we display output probabilities as per the shortest-path expert ($\pi^{\text{teach}}$) for each state. (c/d) Using the same trajectory from (b) we highlight the partial-observations available to the agent (shaded gray) under different filtration function $f^1, f^2$. Notice that, under $f^1$, the agent does not see the goal within its first four steps. The policies $\pi^{\text{IL}}_{f^1}, \pi^{\text{IL}}_{f^2}$, learned by imitating $\pi^{\text{teach}}$, show that imitation results in sub-optimal policies, *i.e.*, $\pi^{\text{IL}}_{f^1}, \pi^{\text{IL}}_{f^2} \neq \pi^{\text{teach}}$.

$f^{\text{teach}}$. For simplicity, we will assume that $\pi^{\text{teach}}_{f^{\text{teach}}}$ is $f^{\text{teach}}$-optimal. Subsequently, we will drop the subscript $f^{\text{teach}}$ unless we wish to explicitly discuss multiple teachers simultaneously.

In IL [45, 52], $\pi_f$ is trained to mimic $\pi^{\text{teach}}$ by minimizing the (expected) cross-entropy between $\pi_f$ and $\pi^{\text{teach}}$ over a set of sampled states $s \in \mathcal{S}$:

$$\min_{\pi_f \in \Pi^{\text{feas.}}_f} \mathbb{E}_\mu[CE(\pi^{\text{teach}}, \pi_f)(S)] \,, \tag{1}$$

where $CE(\pi^{\text{teach}}, \pi_f)(S) = -\pi^{\text{teach}}(S) \odot \log \pi_f(S)$, $\odot$ denotes the usual dot-product, and $S$ is a random variable taking values $s \in \mathcal{S}$ with probability measure $\mu : \mathcal{S} \to [0, 1]$. Often $\mu(s)$ is chosen to equal the frequency with which an exploration policy (*e.g.*, random actions or $\pi^{\text{teach}}$) visits state $s$ in a randomly initialized episode. When it exists, we denote the policy minimizing Eq. (1) as $\pi^{\mu, \pi^{\text{teach}}}_f$. When $\mu$ and $\pi^{\text{teach}}$ are unambiguous, we write $\pi^{\text{IL}}_f = \pi^{\mu, \pi^{\text{teach}}}_f$.

What happens when there is a difference of privilege (or filtration functions) between the teacher and the student? Intuitively, if the information that a teacher uses to make a decision is unavailable to the student then the student has little hope of being able to mimic the teacher's decisions. As we show in our next example, even when optimizing perfectly, depending on the choice of $f$ and $f^{\text{teach}}$, IL may result in $\pi^{\text{IL}}_f$ being uniformly random over a large collection of states. We call the phenomenon that $\pi^{\text{IL}}_f \neq \pi^{\text{teach}}$ the *imitation gap*.

**Example 2** (1D-Lighthouse). We illustrate the imitation gap using a gridworld spanning $\{-N, \ldots, N\}$. The two start states correspond to the goal being at either $-N$ or $N$, while the agent is always initialized at 0 (see Fig. 2a). Clearly, with full state information, $\pi^{\text{teach}}$ maps states to an 'always left' or 'always right' probability distribution, depending on whether the goal is on the left or right, respectively. Suppose now that the agent's visibility is constrained to a radius of $i$ (Fig. 2c shows $i = 1$), *i.e.*, an $f^i$-restricted observation is accessible. An agent following an optimal policy with a visibility of radius $i$ will begin to move deterministically towards any corner, w.l.o.g. assume right. When the agent sees the rightmost edge (from position $N - i$), it will either continue to move right if the goal is visible or, if it's not, move left until it reaches the goal (at $-N$). Now we may ask: what is the best $f^i$-restricted policy that can be learnt by imitating $\pi^{\text{teach}}$ (*i.e.*, what is $\pi^{\text{IL}}_{f^i}$)? *Tragically, the cross-entropy loss causes $\pi^{\text{IL}}_{f^i}$ to be uniform in a large number of states.* In particular, an agent following policy $\pi^{\text{IL}}_{f^i}$ will execute left (and right) actions with probability 0.5, until it is within a distance of $i$ from one of the corners. Subsequently, it will head directly to the goal. See the policies highlighted in Figs. 2c, 2d. The intuition for this result is straightforward: until the agent observes one of the corners it cannot know if the goal is to the right or left and, conditional on its observations, each of these events is equally likely under $\mu$ (assumed uniform). Hence for half of these events the teacher will instruct the agent to go right. For the other half the instruction is to go left. See App. A.1 for a rigorous treatment of this example. In Sec. 4 and Fig. 6, we train $f^i$-restricted policies with $f^j$-optimal teachers for a 2D variant of this example. We empirically verify that a student learns a better policy when imitating teachers whose filtration function is closest to their own.

The above example shows: when a student attempts to imitate an expert that is privileged with information not available to the student, the student learns a version of $\pi^{\text{teach}}$ in which this privileged information is marginalized out. We formalize this intuition in the following proposition.

**Proposition 1** (Policy Averaging).
*In the setting of Section 3.1, suppose that $\Pi^{feas.}$ contains all $f$-restricted policies. Then, for any $s \in \mathcal{S}$ with $o = f(s)$, we have that $\pi_f^{IL}(o) = \mathbb{E}_\mu[\pi^{teach}(S) \mid f(S) = o]$.*

Given our definitions, the proof of this proposition is quite straightforward, see Appendix A.2.

The imitation gap provides theoretical justification for the common practical observation that an agent trained via IL can often be significantly improved by continuing to train the agent using pure RL (*e.g.*, PPO) [38, 13]. Obviously training first with IL and then via pure RL is ad hoc and potentially sub-optimal as discussed in Ex. 1 and empirically shown in Sec. 4. To alleviate this problem, the student should imitate the teacher's policy only in settings where the teacher's policy can, in principle, be exactly reproduced by the student. Otherwise the student should learn via 'standard' RL. To achieve this we introduce ADVISOR.

## 3.2 Adaptive Insubordination (ADVISOR) with Policy Gradients

To close the imitation gap, ADVISOR adaptively weights reward-based and imitation losses. Intuitively, it supervises a student by asking it to imitate a teacher's policy only in those states $s \in \mathcal{S}$ for which the imitation gap is small. For all other states, it trains the student using reward-based RL. To simplify notation, we denote the reward-based RL loss via $\mathbb{E}_\mu[L(\theta, S)]$ for some loss function $L$.[2] This loss formulation is general and spans all policy gradient methods, including A2C and PPO. The imitation loss is the standard cross-entropy loss $\mathbb{E}_\mu[CE(\pi^{\text{teach}}(S), \pi_f(S; \theta))]$. Concretely, the ADVISOR loss is:

$$\mathcal{L}^{\text{ADV}}(\theta) = \mathbb{E}_\mu[w(S) \cdot CE(\pi^{\text{teach}}(S), \pi_f(S; \theta)) + (1 - w(S)) \cdot L(\theta, S)] . \quad (2)$$

Our goal is to find a *weight function* $w : \mathcal{S} \to [0, 1]$ where $w(s) \approx 1$ when the imitation gap is small and $w(s) \approx 0$ otherwise. For this we need an estimator of the distance between $\pi^{\text{teach}}$ and $\pi_f^{\text{IL}}$ at a state $s$ and a mapping from this distance to weights in $[0, 1]$.

We now define a distance estimate $d^0(\pi, \pi_f)(s)$ between a policy $\pi$ and an $f$-restricted policy $\pi_f$ at a state $s$. We can use any common non-negative distance (or divergence) $d$ between probability distributions on $\mathcal{A}$, *e.g.*, in our experiments we use the KL-divergence. While there are many possible strategies for using $d$ to estimate $d^0(\pi, \pi_f)(s)$, perhaps the simplest of these strategies is to define $d^0(\pi, \pi_f)(s) = d(\pi(s), \pi_f(s))$. Note that this quantity does not attempt to use any information about the fiber $f^{-1}(f(s))$ which may be useful in producing more holistic measures of distances.[3] Appendix A.3 considers how those distances can be used in lieu of $d^0$. Next, using the above, we need to estimate the quantity $d^0(\pi^{\text{teach}}, \pi_f^{\text{IL}})(s)$.

Unfortunately it is, in general, impossible to compute $d^0(\pi^{\text{teach}}, \pi_f^{\text{IL}})(s)$ exactly as it is intractable to compute the optimal minimizer $\pi_f^{\text{IL}}$. Instead we leverage an estimator of $\pi_f^{\text{IL}}$ which we term $\pi_f^{\text{aux}}$, and which we will define in the next section.

Given $\pi_f^{\text{aux}}$ we obtain the estimator $d^0(\pi^{\text{teach}}, \pi_f^{\text{aux}})$ of $d^0(\pi^{\text{teach}}, \pi_f^{\text{IL}})$. Additionally, we make use of the monotonically decreasing function $m_\alpha : \mathbb{R}_{\geq 0} \to [0, 1]$, where $\alpha \geq 0$. We define our weight function $w(s)$ for $s \in \mathcal{S}$ as:

$$w(s) = m_\alpha(d^0(\pi^{\text{teach}}, \pi_f^{\text{aux}})(s)) \quad \text{with} \quad m_\alpha(x) = e^{-\alpha x}. \quad (3)$$

---

[2] For readability, we implicitly make three key simplifications. First, computing the expectation $\mathbb{E}_\mu[\ldots]$ is generally intractable, hence we cannot directly minimize losses such as $\mathbb{E}_\mu[L(\theta, S)]$. Instead, we approximate the expectation using rollouts from $\mu$ and optimize the empirical loss. Second, recent RL methods adjust the measure $\mu$ over states as optimization progresses while we assume it to be static for simplicity. Our final simplification regards the degree to which any loss can be, and is, optimized. In general, losses are often optimized by gradient descent and generally no guarantees are given that the global optimum can be found. Extending our presentation to encompass these issues is straightforward but notationally dense.

[3] Measures using such information include $\max_{s' \in f^{-1}(f(s))} d(\pi(s'), \pi_f(s))$ or a corresponding expectation instead of the maximization, *i.e.*, $\mathbb{E}_\mu[d(\pi(S), \pi_f(S)) \mid f(S) = o]$.

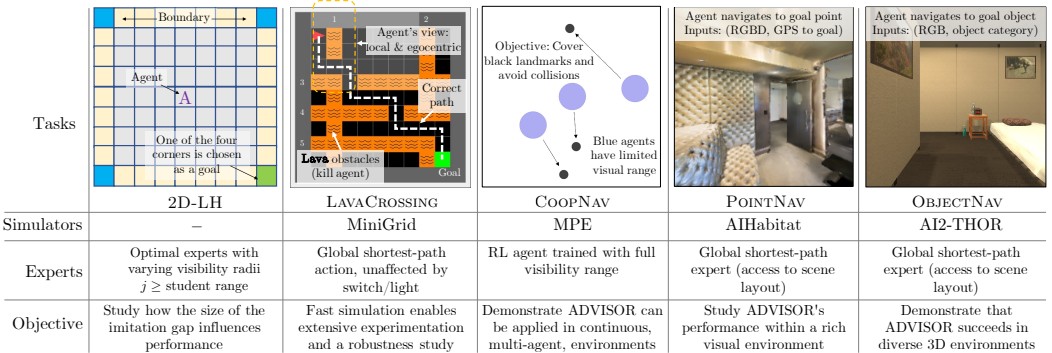

| | | | | |
|---|---|---|---|---|
| Tasks | 2D-LH | LavaCrossing | CoopNav | PointNav | ObjectNav |
| Simulators | – | MiniGrid | MPE | AIHabitat | AI2-THOR |
| Experts | Optimal experts with varying visibility radii $j \geq$ student range | Global shortest-path action, unaffected by switch/light | RL agent trained with full visibility range | Global shortest-path expert (access to scene layout) | Global shortest-path expert (access to scene layout) |
| Objective | Study how the size of the imitation gap influences performance | Fast simulation enables extensive experimentation and a robustness study | Demonstrate ADVISOR can be applied in continuous, multi-agent, environments | Study ADVISOR's performance within a rich visual environment | Demonstrate that ADVISOR succeeds in diverse 3D environments |

Figure 4: **Representative tasks from experiments.** 2D-LH: Harder 2D variant of the gridworld task introduced in Ex. 2. LavaCrossing: one of our 8 tasks in the MiniGrid environment requiring safe navigation. We test up-to $15 \times 15$ grids with 10 lava rivers. CoopNav: A multi-agent cooperative task set in multi-agent particle environments. PointNav: An agent embodied in the AIHabitat environment must navigate using egocentric visual observations to a goal position specified by a GPS coordinate. ObjectNav: An agent in RoboTHOR must navigate to an object of a given category.

### 3.3 The Auxiliary Policy $\pi^{\text{aux}}$: Estimating $\pi_f^{\text{IL}}$ in Practice

In this section we describe how we can, during training, obtain an *auxiliary policy* $\pi_f^{\text{aux}}$ which estimates $\pi_f^{\text{IL}}$. Given this auxiliary policy we estimate $d^0(\pi^{\text{teach}}, \pi_f^{\text{IL}})(s)$ using the plug-in estimator $d^0(\pi^{\text{teach}}, \pi_f^{\text{aux}})(s)$. While plug-in estimators are intuitive and simple to define, they need not be statistically efficient. In Appendix A.4 we consider possible strategies for improving the statistical efficiency of our plug-in estimator via prospective estimation.

In Fig. 3 we provide an overview of how we compute the estimator $\pi_f^{\text{aux}}$ via deep nets. As is common practice [42, 22, 25, 46, 40, 8], the policy net $\pi_f(\cdot; \theta)$ is composed via $a_\nu \circ r_\lambda$ with $\theta = (\nu, \lambda)$, where $a_\nu$ is the *actor head* (possibly complemented in actor-critic models by a *critic head* $v_\nu$) and $r_\lambda$ is called the *representation network*. Generally $a_\nu$ is lightweight, for instance a linear layer or a shallow MLP followed by a soft-max function, while $r_\lambda$ is a deep, and possibly recurrent neural, net. We add another actor head $a_{\nu'}$ to our existing network which shares

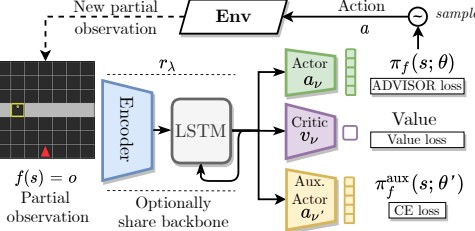

Figure 3: **Model overview.** An auxiliary actor is added and trained only using IL. The 'main' actor policy is trained using the ADVISOR loss.

the underlying representation $r_\lambda$, *i.e.*, $\pi_f^{\text{aux}} = a_{\nu'} \circ r_\lambda$. We experiment with the actors sharing their representation $r_\lambda$ and estimating $\pi_f^{\text{IL}}$ via two separate networks, *i.e.*, $\theta' = (\nu', \lambda')$. In practice we train $\pi_f(\cdot; \theta)$ and $\pi_f^{\text{aux}}(\cdot; \theta)$ jointly using stochastic gradient descent, as summarized in Alg. A.1.

## 4 Experiments

We rigorously compare ADVISOR to IL methods, RL methods, and popularly-adopted (but often ad hoc) IL & RL combinations. In particular, we evaluate 15 learning methods. We do this over thirteen tasks – realizations of Ex. 1 & Ex. 2, eight tasks of varying complexity within the fast, versatile MiniGrid environment [8, 9], Cooperative Navigation (CoopNav) with reduced visible range in the multi-agent particle environment (MPE) [43, 37], PointGoal navigation (PointNav) using the Gibson dataset in AIHabitat [71, 54], and ObjectGoal Navigation (ObjectNav) in RoboTHOR [14].[4] Furthermore, to probe robustness, we train 50 hyperparameter variants for each of the 15

---

[4]The RoboTHOR environment is a sub-environment of AI2-THOR [31].

| Tasks → | PD | LAVACROSSING | | | | WALLCROSSING | | | |
|---|---|---|---|---|---|---|---|---|---|
| Training routines ↓ | - | Base Ver. | Corrupt Exp. | Faulty Switch | Once Switch | Base Ver. | Corrupt Exp. | Faulty Switch | Once Switch |
| RL | 0 | 0 | 0 | 0.01 | 0 | 0.09 | 0.07 | 0.12 | 0.05 |
| IL | -0.59 | 0.88 | 0.02 | 0.02 | 0 | 0.96 | 0.05 | 0.17 | 0.11 |
| IL & RL | -0.17 | 0.94 | 0.74 | 0.04 | 0 | **0.97** | 0.18 | 0.17 | 0.1 |
| Demo. Based | -0.09 | **0.96** | 0.2 | 0.02 | 0 | **0.97** | 0.07 | 0.18 | 0.11 |
| ADV. Based (ours) | **1** | **0.96** | **0.94** | **0.77** | **0.8** | **0.97** | **0.31** | **0.38** | **0.45** |

Table 1: **Expected rewards for the POISONEDDOORS task and MINIGRID tasks.** For each of our 15 training routines we report the expected maximum validation set performance (when given a budget of 10 random hyperparameter evaluations) after training for ≈300k steps in POISONEDDOORS and ≈1Mn steps in our 8 MINIGRID tasks. The maximum reward is 1 for the MINIGRID tasks.

learning methods for our MINIGRID tasks. We find ADVISOR-based methods outperform or match performance of all baselines.

All code to reproduce our experiments will be made public under the Apache 2.0 license.[5] The environments used are public for academic and commercial use under the Apache 2.0 (MINIGRID and ROBOTHOR) and MIT licence (MPE and AIHABITAT).

## 4.1 Tasks

Detailed descriptions of our tasks (and teachers) are deferred to Appendix A.5. See Fig. 4 for a high-level overview of 5 representative tasks.

## 4.2 Baselines and ADVISOR-based Methods

We briefly introduce baselines and variants of our ADVISOR method. Further details of all methods are in Appendix A.7. For fairness, the same model architecture is shared across all methods (recall Fig. 3, Sec. 3.3). We defer implementation details to Appendix A.8.

- **RL only.** Proximal Policy Optimization [58] serves as the pure RL baseline for all our tasks with a discrete action space. For the continuous and multi-agent COOPNAV task, we follow prior work and adopt MADDPG [37, 36].

- **IL only.** IL baselines where supervision comes from an expert policy with different levels of teacher-forcing (tf), *i.e.*, tf=0, tf annealed from 1→0, and tf=1. This leads to Behaviour Cloning (BC), Data Aggregation (DAgger or †), and $BC^{tf=1}$, respectively [53, 2, 52].

- **IL & RL.** Baselines that use a mix of IL and RL losses, either in sequence or in parallel. These are popularly adopted in the literature to warm-start agent policies. Sequential combinations include BC then PPO (BC→PPO), DAgger then PPO († → PPO), and $BC^{tf=1}$ → PPO. The parallel combination of BC + PPO(static) is a static analog of our adaptive combination of IL and RL losses.

- **Demonstration-based.** These agents imitate expert demonstrations and hence get no supervision beyond the states in the demonstrations. We implement $BC^{demo}$, its combination with PPO ($BC^{demo}$ + PPO), and Generative Adversarial Imitation Learning (GAIL) [24].

- **ADVISOR-based (ours).** Our Adaptive Insubordination methodology can learn from an expert policy and can be given a warm-start via BC or DAgger. This leads to ADVISOR (ADV), $BC^{tf=1}$ → ADV, and † → ADV) baselines. Similarly, $ADV^{demo}$ + PPO employs Adaptive Insubordination to learn from expert demonstrations while training with PPO on on-policy rollouts.

## 4.3 Evaluation

**Fair Hyperparameter Tuning.** Often unintentionally done, extensively tuning the hyperparameters (hps) of a proposed method and not those of the baselines can introduce unfair bias into evaluations. We avoid this by considering two strategies. For PD and all MINIGRID tasks, we follow recent best practices [15]. Namely, we tune each method by randomly sampling a fixed number of hps and

---

[5]See https://unnat.github.io/advisor/ for an up-to-date link to this code.

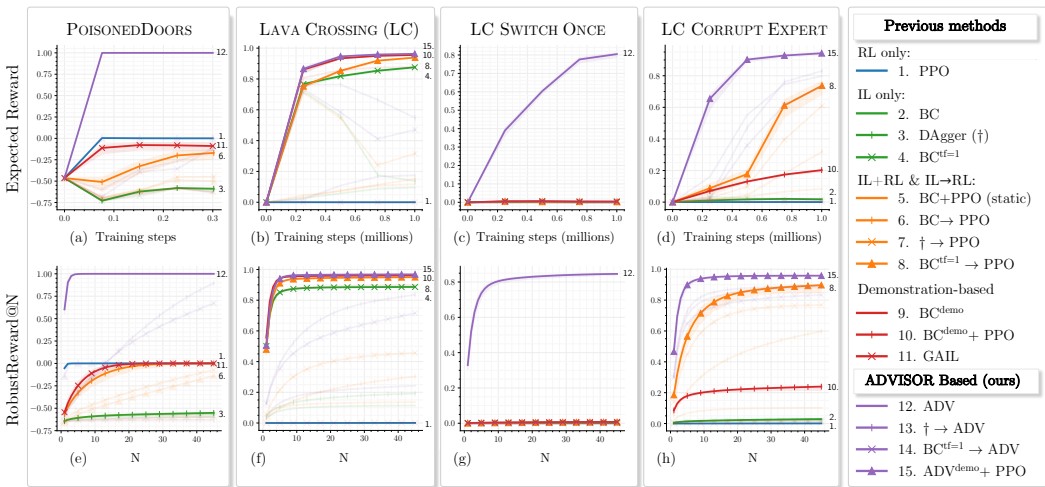

Figure 5: **Evaluation following [15].** Plots for 15 training routines in four selected tasks (additional plots in appendix). For clarity, we highlight the best performing training routine within five categories, *e.g.*, RL only, IL only *etc.* (details in Sec. 4.2) with all other plots shaded lighter. (a)-(d) As described in Sec. 4.3 we plot *RobustReward*@10 at multiple points during training. (e)-(f) Plots of *RobustReward*@$N$ for values of $N \in \{1, \ldots, 45\}$. Recall that *RobustReward*@$N$ is the expected validation reward of best-found model when allowed $N$ random hyperparameter evaluations.

| Tasks → | PointGoal Navigation | | | | ObjectGoal Navigation | | | | Cooperative Navigation | |
| | SPL | | Success | | SPL | | Success | | Reward | |
| Training routines ↓ | @10% | @100% | @10% | @100% | @10% | @100% | @10% | @100% | @10% | @100% |
|---|---|---|---|---|---|---|---|---|---|---|
| RL only | 30.9 | 54.7 | 54.7 | 79.0 | 6.7 | 13.1 | 11.1 | **31.6** | −561.8 | −456.0 |
| IL only | 30.1 | 68.7 | 35.5 | 76.7 | 3.8 | 9 | 8.8 | 13.6 | −460.3 | −416.7 |
| IL + RL static | 48.9 | 71.5 | 56.7 | 78.2 | 6.5 | 11.3 | 11.7 | 19.8 | −475.5 | −424.6 |
| ADVISOR (ours) | **57.7** | **77.1** | **67.3** | **88.2** | **11.9** | **14.1** | **22.7** | 29.9 | **−419.9** | **−405.6** |

Table 2: **Quantitative results for high-fidelity visual environments and continuous control.** Validation set performance after 10% and 100% of training has completed for four training routines on the POINTNAV, OBJECTNAV, and COOPNAV tasks (specifics of these routines can be found in the Appendix). For POINTNAV and OBJECTNAV we include the common success weighted path length (SPL) metric [1] in addition to the success rate.

reporting, for each baseline, an estimate of

$$RobustReward@K = \mathbb{E}[\text{Val. reward of best model from } k \text{ random hyperparam. evaluations}] \quad (4)$$

for $1 \leq k \leq 45$. For this we must train 50 models per method, *i.e.*, 750 for each of these nine tasks. In order to show learning curves over training steps we also report *RobustReward*@10 at 5 points during training. More details in Appendix A.9. For 2D-LH, we tune the hps of a competing method and use these hps for all other methods.

**Training.** For the eight MINIGRID tasks, we train each of the 50 training runs for 1 million steps. For 2D-LH/PD, models saturate much before $3 \cdot 10^5$ steps. POINTNAV, OBJECTNAV, and COOPNAV are trained for standard budgets of 50Mn, 100Mn, and 1.5Mn steps. Details are in Appendix A.10.

**Metrics.** We record standard metrics for each task. This includes avg. rewards (PD, MINIGRID tasks, and OBJECTNAV), and avg. episode lengths (2D-LH). Following visual navigation works [1, 54, 14], we report success rates and success-weighted path length (SPL) for POINTNAV and OBJECTNAV. In the following, we report a subset of the above and defer additional plots to Appendix A.11.

### 4.4 Results

In the following, we include takeaways based on the results in Fig. 5, Fig. 6, Tab. 1, and Tab. 2.

**Smaller imitation gap $\implies$ better performance.** A central claim of our paper is that the imitation gap is not merely a theoretical concern: the degree to which the teacher is privileged over the student has significant impact on the student's performance. To study this empirically, we vary the degree to which teachers are privileged over its students in our 2D-LH

task. In particular, we use behavior cloning to train an $f^i$-restricted policy (*i.e.*, an agent that can see $i$ grid locations away) using an $f^j$-optimal teacher 25 times. Each policy is then evaluated on 200 random episodes and the average episode length (lower being better) is recorded. For select $i, j$ pairs we show boxplots of the 25 average episode lengths in Fig. 6. See our appendix for similar plots when using other training routines (*e.g.*, ADVISOR).

Grey vertical lines show optimal average episode lengths for $f^i$-restricted policies. We find that training an $f^i$-restricted policy with an $f^j$-expert results in a near optimal policy when $i = j$ but even small increases in $j$ dramatically decrease performance. While performance tends to drop with increasing $j$, the largest $i, j$ gaps do not consistently correspond to the worst performing models. While this seems to differ from our results in Ex. 2, recall that there the policy $\mu$ was fixed while here it varies through training, resulting in complex learning dynamics. Surprisingly we also find that, even when there is no imitation gap (*e.g.*, the $i = j$ case), ADVISOR can outperform BC, see App. A.6.

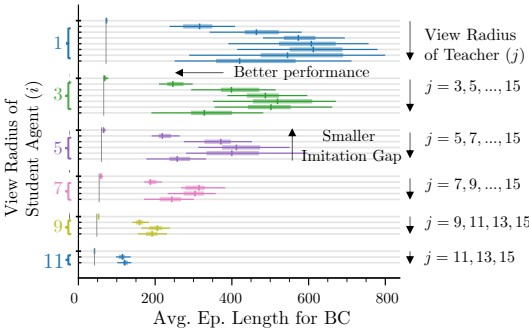

Figure 6: The size of the imitation gap directly impacts performance (in 2D-LH).

**ADVISOR outperforms, even in complex visual environments.** Across all of our tasks, ADVISOR-based methods perform as well or better than competing methods. In particular, see Tab. 1 for our results on the POISONEDDOORS (PD) and MINIGRID tasks and Tab. 2 for our results on the POINTNAV, OBJECTNAV, and COOPNAV tasks. 2D-LH results are deferred to the Appendix.

While the strong performance of ADVISOR is likely expected on our PD, MINIGRID, and 2D-LH tasks (indeed we designed a subset of these with the explicit purpose of studying the imitation gap), it is nonetheless surprising to see that in the PD and LC ONCE SWITCH tasks, all non-ADVISOR methods completely fail. Moreover, it is extremely promising to see that ADVISOR can provide substantial benefits in a variety of standard tasks, namely OBJECTNAV, POINTNAV, and COOPNAV with limited visible range. Note that OBJECTNAV and POINTNAV are set in 3D high-fidelity visual environments while COOPNAV requires multi-agent collaboration in a continuous space.

**ADVISOR is sample efficient.** To understand the sample efficiency of ADVISOR, we plot validation set performance over training of select tasks (see Figures 5a to 5d) and, in Table 2 we show performance of our models after 10% of training has elapsed for the OBJECTNAV, POINTNAV, and COOPNAV tasks. Note that in Table 2, ADVISOR trained models frequently reach better performance after 10% of training than other methods manage to reach by the end of training.

**ADVISOR is robust.** Rigorously studying sensitivity to hyperparameter choice requires retraining every method under consideration tens to hundreds of times. This computational task can make evaluating our methods on certain tasks infeasible (training a single POINTNAV or OBJECTNAV model can easily require a GPU-week of computation). Because of these computational constraints, we limit our study of robustness to the PD and MINIGRID tasks. In Figures 5e to 5h (additional results in Appendix) we plot, for each of the 15 evaluated methods, how the expected performance of each method behaves as we increase the budget of random hyperparameter evaluations. In general, relatively few hyperparameter evaluations are required for ADVISOR before a high performance model is expected to be found.

**Expert demonstrations can be critical to success.** While it is frequently assumed that on-policy expert supervision is better than learning from off-policy demonstrations, we found several instances in our MINIGRID experiments where demonstration-based methods outperformed competing methods. See, for example, Figures 5b and 5f. In such cases our demonstration-based ADVISOR variant (see Appendix A.7 for details) performed very well.

**ADVISOR helps even when the expert is corrupted.** In LC CORRUPT EXPERT and WC CORRUPT EXPERT, where the expert is designed to be corrupted (outputting random actions as supervision) when the agent gets sufficiently close to the goal. While ADVISOR was not designed with the possibility of corrupted experts in mind, Figures 5d and 5h (see also Table 1) show that ADVISOR can succeed despite this corruption.

# 5 Conclusion

We propose the *imitation gap* as one explanation for the empirical observation that imitating "more intelligent" teachers can lead to worse policies. While prior work has, implicitly, attempted to bridge this gap, we introduce a principled adaptive weighting technique (ADVISOR), which we test on a suite of thirteen tasks. Due to the fast rendering speed of MINIGRID, PD, and 2D-LH, we could undertake a study where we trained over 6 billion steps, to draw statistically significant inferences.

# 6 Limitations and Societal Impact

While we have attempted to robustly evaluate our proposed ADVISOR methodology, we have primarily focused our experiments on navigational tasks where shortest path experts can be quickly computed. Further work is needed to validate that ADVISOR can be successful in other domains, *e.g.*, imitation in interactive robotic tasks or natural language applications.

While the potential for direct negative societal impact of this work is small, it is worth noting that, in enabling agents to learn more effectively from expert supervision, this work makes imitation learning a more attractive option to RL researchers. If expert supervision is obtained from humans, RL agents trained with such data will inevitably reproduce any (potentially harmful) biases of these humans.

## Acknowledgements

This material is based upon work supported in part by the National Science Foundation under Grants No. 1563727, 1718221, 1637479, 165205, 1703166, 2008387, 2045586, 2106825, MRI #1725729, NIFA award 2020-67021-32799, Samsung, 3M, Sloan Fellowship, NVIDIA Artificial Intelligence Lab, Allen Institute for AI, Amazon, AWS Research Awards, and Siebel Scholars Award. We thank Nan Jiang and Tanmay Gangwani for feedback on this work.

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
