# Appendix: Bridging the Imitation Gap by Adaptive Insubordination

The appendix includes theoretical extensions of ideas presented in the main paper and details of empirical analysis. We structure the appendix into the following subsections:

## A  Additional Information

### A.1  Formal treatment of Example 2

Let $N \geq 1$ and consider a 1-dimensional grid-world with states $\mathcal{S} = \{-N, N\} \times \{0, \ldots, T\} \times \{-N, \ldots, N\}^T$. Here $g \in \{-N, N\}$ are possible goal positions, elements $t \in \{0, \ldots, T\}$ correspond to the episode's current timestep, and $(p_i)_{i=1}^T \in \{-N, \ldots, N\}^T$ correspond to possible agent trajectories of length $T$. Taking action $a \in \mathcal{A} = \{\text{left}, \text{right}\} = \{-1, 1\}$ in state $(g, t, (p_i)_{i=1}^T) \in \mathcal{S}$ results in the deterministic transition to state $(g, t+1, (p_1, \ldots, p_t, \text{clip}(p_t + a, -N, N), 0, \ldots, 0))$. An episode start state is chosen uniformly at random from the set $\{(\pm N, 0, (0, \ldots, 0))\}$ and the goal of the agent is to reach some state $(g, t, (p_i)_{i=1}^T)$ with $p_t = g$ in the fewest steps possible. We now consider a collection of filtration functions $f^i$, that allow the agent to see spaces up to $i$ steps left/right of its current position but otherwise has perfect memory of its actions. See Figs. 2c, 2d for examples of $f^1$- and $f^2$-restricted observations. For $0 \leq i \leq N$ we define $f^i$ so that

$$f^i(g, t, (p_i)_{i=1}^T) = ((\ell_0, \ldots, \ell_t), (p_1 - p_0, \ldots, p_t - p_{t-1})) \quad \text{and} \tag{5}$$

$$\ell_j = (1_{[p_j + k = N]} - 1_{[p_j + k = -N]} \mid k \in \{-i, \ldots, i\}) \quad \text{for } 0 \leq j \leq t. \tag{6}$$

Here $\ell_j$ is a tuple of length $2 \cdot i + 1$ and corresponds to the agent's view at timestep $j$ while $p_{k+1} - p_k$ uniquely identifies the action taken by the agent at timestep $k$. Let $\pi^{\text{teach}}$ be the optimal policy given full state information so that $\pi^{\text{teach}}(g, t, (p_i)_{i=1}^T) = (1_{[g=-N]}, 1_{[g=N]})$ and let $\mu$ be a uniform distribution over states in $\mathcal{S}$. It is straightforward to show that an agent following policy $\pi_{f^i}^{\text{IL}}$ will take random actions until it is within a distance of $i$ from one of the corners $\{-N, N\}$ after which it will head directly to the goal, see the policies highlighted in Figs. 2c, 2d. The intuition for this result is straightforward: until the agent observes one of the corners it cannot know if the goal is to the right or left and, conditional on its observations, each of these events is equally likely under $\mu$. Hence in half of these events the expert will instruct the agent to go right and in the other half

---

[6]We overload main paper's notation $d^0(\pi, \pi_f)(s)$ with $d_\pi^0(\pi_f)(s)$

**Algorithm A.1: On-policy ADVISOR algorithm overview.** Some details omitted for clarity.

**Input:** Trainable policies $(\pi_f, \pi_f^{\text{aux}})$, expert policy $\pi^{\text{teach}}$, rollout length $L$, environment $\mathcal{E}$.
**Output:** Trained policy

1 **begin**
2    Initialize the environment $\mathcal{E}$
3    $\theta \leftarrow$ randomly initialized parameters
4    **while** *Training completion criterion not met* **do**
5      Take $L$ steps in the environment using $\pi_f(\cdot; \theta)$ and record resulting rewards and
       observations (restarting $\mathcal{E}$ whenever the agent has reached a terminal state)
6      Evaluate $\pi_f^{\text{aux}}(\cdot; \theta)$ and $\pi^{\text{teach}}$ at each of the above steps
7      $L \leftarrow$ the empirical version of the loss from Eq. (2) computed using the above rollout
8      Compute $\nabla_\theta L$ using backpropagation
9      Update $\theta$ using $\nabla_\theta L$ via gradient descent
10    **return** $\pi_f(\cdot; \theta)$

to go left. The cross entropy loss will thus force $\pi_{fi}^{\text{IL}}$ to be uniform in all such states. Formally, we will have, for $s = (g, t, (p_i)_{i=1}^T)$, $\pi_{fi}^{\text{IL}}(s) = \pi^{\text{teach}}(s)$ if and only if $\min_{0 \leq q \leq t}(p_q) - i \leq -N$ or $\max_{0 \leq q \leq t}(p_q) + i \geq N$ and, for all other $s$, we have $\pi_{fi}^{\text{IL}}(s) = (1/2, 1/2)$. In Sec. 4, see also Fig. 6, we train $f^i$-restricted policies with $f^j$-optimal teachers for a 2D variant of this example. ∎

## A.2 Proof of Proposition 1

We wish to show that the minimizer of $\mathbb{E}_\mu[-\pi_{f^e}^{\text{teach}}(S) \odot \log \pi_f(S)]$ among all $f$-restricted policies $\pi_f$ is the policy $\overline{\pi} = \mathbb{E}_\mu[\pi^{\text{teach}}(S) \mid f(S)]$. This is straightforward, by the law of iterated expectations and as $\pi_f(s) = \pi_f(f(s))$ by definition. We obtain

$$
\begin{aligned}
\mathbb{E}_\mu[-\pi_{f^e}^{\text{teach}}(S) \odot \log \pi_f(S)] &= -\mathbb{E}_\mu[E_\mu[\pi_{f^e}^{\text{teach}}(S) \odot \log \pi_f(S) \mid f(S)]] \\
&= -\mathbb{E}_\mu[E_\mu[\pi_{f^e}^{\text{teach}}(S) \odot \log \pi_f(f(S)) \mid f(S)]] \\
&= -\mathbb{E}_\mu[E_\mu[\pi_{f^e}^{\text{teach}}(S) \mid f(S)] \odot \log \pi_f(f(S))] \\
&= \mathbb{E}_\mu[-\overline{\pi}(f(S)) \odot \log \pi_f(f(S))] .
\end{aligned}
\tag{7}
$$

Now let $s \in \mathcal{S}$ and let $o = f(s)$. It is well known, by Gibbs' inequality, that $-\overline{\pi}(o) \odot \log \pi_f(o)$ is minimized (in $\pi_f(o)$) by letting $\pi_f(o) = \overline{\pi}(o)$ and this minimizer is feasible as we have assumed that $\Pi_f$ contains *all* $f$-restricted policies. Hence it follows immediately that Eq. (7) is minimized by letting $\pi_f = \overline{\pi}$ which proves the claimed proposition.

## A.3 Other Distance Measures

As discussed in Section 3.2, there are several different choices one may make when choosing a measure of distance between the expert policy $\pi^{\text{teach}}$ and an $f$-restricted policy $\pi_f$ at a state $s \in \mathcal{S}$. The measure of distance we use in our experiments, $d_{\pi^{\text{teach}}}^0(\pi_f)(s) = d(\pi^{\text{teach}}(s), \pi_f(s))$, has the (potentially) undesirable property that $f(s) = f(s')$ does not imply that $d_{\pi^{\text{teach}}}^0(\pi_f)(s) = d_{\pi^{\text{teach}}}^0(\pi_f)(s')$. While an in-depth evaluation of the merits of different distance measures is beyond this current work, we suspect that a careful choice of such a distance measure may have a substantial impact on the speed of training. The following proposition lists a collection of possible distance measures with a conceptual illustration given in Fig. A.1.

**Proposition 2.** *Let $s \in \mathcal{S}$ and $o = f(s)$ and for any $0 < \beta < \infty$ define, for any policy $\pi$ and $f$-restricted policy $\pi_f$,*

$$
d_{\mu,\pi}^\beta(\pi_f)(s) = E_\mu[(d_\pi^0(\pi_f)(S))^\beta \mid f(S) = f(s)]^{1/\beta},
\tag{8}
$$

*with $d_{\mu,\pi}^\infty(\pi_f)(s)$ equalling the essential supremum of $d_\pi^0(\pi_f)$ under the conditional distribution $P_\mu(\cdot \mid f(S) = f(s))$. As a special case note that*

$$
d_{\mu,\pi}^1(\pi_f)(s) = E_\mu[d_\pi^0(\pi_f)(S) \mid f(S) = f(s)].
$$

*Then, for all $\beta \geq 0$ and $s \in \mathcal{S}$ (almost surely $\mu$), we have that $\pi(s) \neq \pi_f(f(s))$ if and only if $d_\pi^\beta(\pi_f)(s) > 0$.*

*Proof.* This statement follows trivially from the definition of $\pi^{\text{IL}}$ and the fact that $d(\pi, \pi') \geq 0$ with $d(\pi, \pi') = 0$ if and only if $\pi = \pi'$. $\qquad\square$

The above proposition shows that any $d^\beta$ can be used to consistently detect differences between $\pi^{\text{teach}}$ and $\pi_f^{\text{IL}}$, *i.e.*, it can be used to detect the imitation gap. Notice also that for any $\beta > 0$ we have that $d_{\mu,\pi^{\text{teach}}}^\beta(\pi_f^{\text{IL}})(s) = d_{\mu,\pi^{\text{teach}}}^\beta(\pi_f^{\text{IL}})(s')$ whenever $f(s) = f(s')$.

As an alternative to using $d^0$, we now describe how $d_{\mu,\pi^{\text{teach}}}^1(\pi_f^{\text{IL}})(s)$ can be estimated in practice during training. Let $\pi_f^{\text{aux}}$ be an estimator of $\pi_f^{\text{IL}}$ as usual. To estimate $d_{\mu,\pi^{\text{teach}}}^1(\pi_f^{\text{IL}})(s)$ we assume we have access to a function approximator $g_\psi : \mathcal{O}_f \to \mathbb{R}$ parameterized by $\psi \in \Psi$, *e.g.*, a neural network. Then we estimate $d_{\mu,\pi^{\text{teach}}}^1(\pi_f^{\text{IL}})(s)$ with $g_{\widehat\psi}$ where $\widehat\psi$ is taken to be the minimizer of the loss

$$\mathcal{L}_{\mu,\pi^{\text{teach}},\pi_f^{\text{aux}}}(\psi) = E_\mu\left[\left(d(\pi^{\text{teach}}(S), \pi_f^{\text{aux}}(f(S))) - g_\psi(f(S))\right)^2\right]. \tag{9}$$

The following proposition then shows that, assuming that $d_{\mu,\pi^{\text{teach}}}^1(\pi_f^{\text{aux}}) \in \{g_\psi \mid \psi \in \Psi\}$, $g_{\widehat\psi}$ will equal $d_{\mu,\pi^{\text{teach}}}^1(\pi_f^{\text{aux}})$ and thus $g_{\widehat\psi}$ may be interpreted as a plug-in estimator of $d_{\mu,\pi^{\text{teach}}}^1(\pi_f^{\text{IL}})$.

**Proposition 3.** *For any $\psi \in \Psi$,*

$$\mathcal{L}_{\mu,\pi^{teach},\pi_f^{aux}}(\psi) = E_\mu[(d_{\mu,\pi^{teach}}^1(\pi_f^{aux})(S) - g_\psi(f(S)))^2] + c,$$

*where $c = E_\mu[(d(\pi^{teach}(S), \pi^{aux}(f(S))) - d_{\mu,\pi^{teach},\widehat{}}^1(S))^2]$ is constant in $\psi$ and this implies that if $d_{\mu,\pi^{teach}}^1(\pi_f^{aux}) \in \{g_\psi \mid \psi \in \Psi\}$ then $g_{\widehat\psi} = d_{\mu,\pi^{teach}}^1(\pi_f^{aux})$.*

*Proof.* In the following we let $O_f = f(S)$. We now have that

$$E_\mu[\left(d(\pi^{\text{teach}}(S), \pi_f^{\text{aux}}(O_f)) - g_\psi(O_f)\right)^2]$$
$$= E_\mu[((d(\pi^{\text{teach}}(S), \pi_f^{\text{aux}}(O_f)) - d_{\mu,\pi^{\text{teach}}}^1(\pi_f^{\text{aux}})(S)) + (d_{\mu,\pi^{\text{teach}}}^1(\pi_f^{\text{aux}})(S) - g_\psi(O_f)))^2]$$
$$= E_\mu[(d(\pi^{\text{teach}}(S), \pi_f^{\text{aux}}(O_f)) - d_{\mu,\pi^{\text{teach}}}^1(\pi_f^{\text{aux}})(S))^2] + E_\mu[(d_{\mu,\pi^{\text{teach}}}^1(\pi_f^{\text{aux}})(S) - g_\psi(O_f)))^2]$$
$$\quad + 2 \cdot E_\mu[((d(\pi^{\text{teach}}(S), \pi_f^{\text{aux}}(O_f)) - d_{\mu,\pi^{\text{teach}}}^1(\pi_f^{\text{aux}})(S)) \cdot (d_{\mu,\pi^{\text{teach}}}^1(\pi_f^{\text{aux}})(S) - g_\psi(O_f)))]$$
$$= c + E_\mu[(d_{\mu,\pi^{\text{teach}}}^1(\pi_f^{\text{aux}})(S) - g_\psi(O_f)))^2]$$
$$\quad + 2 \cdot E_\mu[((d(\pi^{\text{teach}}(S), \pi_f^{\text{aux}}(O_f)) - d_{\mu,\pi^{\text{teach}}}^1(\pi_f^{\text{aux}})(S)) \cdot (d_{\mu,\pi^{\text{teach}}}^1(\pi_f^{\text{aux}})(S) - g_\psi(O_f)))].$$

Now as as $d_{\mu,\pi^{\text{teach}}}^1(\pi_f^{\text{aux}})(s) = d_{\mu,\pi^{\text{teach}}}^1(\pi_f^{\text{aux}})(s')$ for any $s, s'$ with $f(s) = f(s')$ we have that $d_{\mu,\pi^{\text{teach}}}^1(\pi_f^{\text{aux}})(S) - g_\psi(O_f)$ is constant conditional on $O_f$ and thus

$$E_\mu[(d(\pi^{\text{teach}}(S), \pi_f^{\text{aux}}(O_f)) - d_{\mu,\pi^{\text{teach}}}^1(\pi_f^{\text{aux}})(S)) \cdot (d_{\mu,\pi^{\text{teach}}}^1(\pi_f^{\text{aux}})(S) - g_\psi(O_f)) \mid O_f]$$
$$= E_\mu[(d(\pi^{\text{teach}}(S), \pi_f^{\text{aux}}(O_f)) - d_{\mu,\pi^{\text{teach}}}^1(\pi_f^{\text{aux}})(S) \mid O_f] \cdot E_\mu[d_{\mu,\pi^{\text{teach}}}^1(\pi_f^{\text{aux}})(S) - g_\psi(O_f) \mid O_f]$$
$$= E_\mu[d_{\mu,\pi^{\text{teach}}}^1(\pi_f^{\text{aux}})(S) - d_{\mu,\pi^{\text{teach}}}^1(\pi_f^{\text{aux}})(S) \mid O_f] \cdot E_\mu[d_{\mu,\pi^{\text{teach}}}^1(\pi_f^{\text{aux}})(S) - g_\psi(O_f) \mid O_f]$$
$$= 0.$$

Combining the above results and using the law of iterated expectations gives the desired result. $\quad\square$

## A.4   Future Directions in Improving Distance Estimators

In this section we highlight possible directions towards improving the estimation of $d_{\pi^{\text{teach}}}^0(\pi_f^{\text{IL}})(s)$ for $s \in \mathcal{S}$. As a comprehensive study of these directions is beyond the scope of this work, our aim in this section is intuition over formality. We will focus on $d^0$ here but similar ideas can be extended to other distance measures, *e.g.*, those in Sec. A.3.

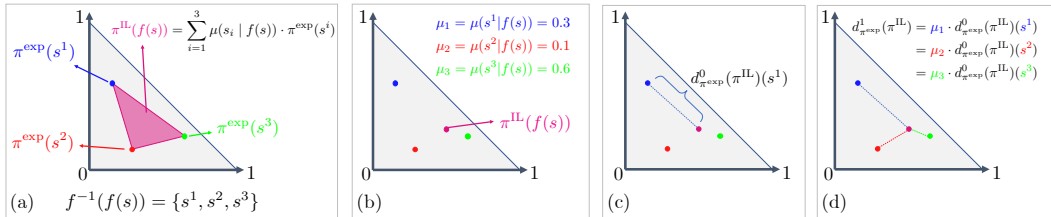

Figure A.1: **Concept Illustration.** Here we illustrate several of the concepts from our paper. Suppose our action space $\mathcal{A}$ contains three elements. Then for any $s \in \mathcal{S}$ and policy $\pi$, the value $\pi(s)$ can be represented as a single point in the 2-dimensional probability simplex $\{(x, y) \in \mathbb{R}^2 \mid x \geq 0, y \geq 0, x + y \leq 1\}$ shown as the grey area in (a). Suppose that the fiber $f^{-1}(f)$ contains the three unique states $s^1$, $s^2$, and $s^3$. In (a) we show the hypothetical values of $\pi^{\text{exp}}$ when evaluated at these points. Proposition 1 says that $\pi^{\text{IL}}(s)$ lies in the convex hull of $\{\pi^{\text{teach}}(s^i)\}_{i=1}^3$ visualized as a magenta triangle in (a). Exactly where $\pi^{\text{IL}}(s)$ lies depends on the probability measure $\mu$, in (b) we show how a particular instantiation of $\mu$ may result in a realization of $\pi^{\text{IL}}(s)$ (not to scale). (c) shows how $d^1_{\pi^{\text{teach}}}$ measures the distance between $\pi^{\text{teach}}(s^1)$ and $\pi^{\text{IL}}(s^1)$. Notice that it ignores $s^2$ and $s^3$. In (d), we illustrate how $d^0_{\pi^{\text{teach}}}$ produces a "smoothed" measure of distance incorporating information about all $s^i$.

As discussed in the main paper, we estimate $d^0_{\pi^{\text{teach}}}(\pi^{\text{IL}}_f)(s)$ by first estimating $\pi^{\text{IL}}_f$ with $\pi^{\text{aux}}_f$ and then forming the "plug-in" estimator $d^0_{\pi^{\text{teach}}}(\pi^{\text{aux}}_f)(s)$. For brevity, we will write $d^0_{\pi^{\text{teach}}}(\pi^{\text{aux}}_f)(s)$ as $\widehat{d}$. While such plug-in estimators are easy to estimate and conceptually compelling, they need not be statistically efficient. Intuitively, the reason for this behavior is because we are spending too much effort in trying to create a high quality estimate $\pi^{\text{aux}}_f$ of $\pi^{\text{IL}}_f$ when we should be willing to sacrifice some of this quality in service of obtaining a better estimate of $d^0_{\pi^{\text{teach}}}(\pi^{\text{IL}}_f)(s)$. Very general work in this area has brought about the targeted maximum-likelihood estimation (TMLE) [63] framework. Similar ideas may be fruitful in improving our estimator $\widehat{d}$.

Another weakness of $\widehat{d}$ discussed in the main paper is that is not prospective. In the main paper we assume, for readability, that we have trained the estimator $\pi^{\text{aux}}_f$ before we train our main policy. In practice, we train $\pi^{\text{aux}}_f$ alongside our main policy. Thus the quality of $\pi^{\text{aux}}_f$ will improve throughout training. To clarify, suppose that, for $t \in [0, 1]$, $\pi^{\text{aux}}_{f,t}$ is our estimate of $\pi^{\text{IL}}_f$ after $(100 \cdot t)\%$ of training has completed. Now suppose that $(100 \cdot t)\%$ of training has completed and we wish to update our main policy using the ADVISOR loss given in Eq. (2). In our current approach we estimate $d^0_{\pi^{\text{teach}}}(\pi^{\text{IL}}_f)(s)$ using $d^0_{\pi^{\text{teach}}}(\pi^{\text{aux}}_{f,t})(s)$ when, ideally, we would prefer to use $d^0_{\pi^{\text{teach}}}(\pi^{\text{aux}}_{f,1})(s)$ from the end of training. Of course we will not know the value of $d^0_{\pi^{\text{teach}}}(\pi^{\text{aux}}_{f,1})(s)$ until the end of training but we can, in principle, use time-series methods to estimate it. To this end, let $q_\omega$ be a time-series model with parameters $\omega \in \Omega$ (*e.g.*, $q_\omega$ might be a recurrent neural network) and suppose that we have stored the model checkpoints $(\pi^{\text{aux}}_{f,i/K} \mid i/K \leq t)$. We can then train $q_\omega$ to perform forward prediction, for instance to minimize

$$\sum_{j=1}^{\lfloor t \cdot K \rfloor} \left( d^0_{\pi^{\text{teach}}}(\pi^{\text{aux}}_{f,j/K})(s) - q_\omega\big(s, (\pi^{\text{aux}}_{f,i/K}(s))_{i=1}^{j-1}\big) \right)^2 ,$$

and then use this trained $q_\omega$ to predict the value of $d^0_{\pi^{\text{teach}}}(\pi^{\text{aux}}_{f,1})(s)$. The advantage of this prospective estimator $q_\omega$ is that it can detect that the auxiliary policy will eventually succeed in exactly imitating the expert in a given state and thus allow for supervising the main policy with the expert cross entropy loss earlier in training. The downside of such a method: it is significantly more complicated to implement and requires running inference using saved model checkpoints.

## A.5 Additional Task Details

In Figure 4 we gave a quick qualitative glimpse at the various tasks we use in our experiments. Here, we provide additional details for each of them along with information about observation space associated with each task. For training details for the tasks, please see Sec. A.10.

### A.5.1 PoisonedDoors (PD)

This environment is a reproduction of our example from Sec. 1. An agent is presented with $N = 4$ doors $d_1, \ldots, d_4$. Door $d_1$ is locked, requiring a fixed $\{0, 1, 2\}^{10}$ code to open, but always results in a reward of 1 when opened. For some randomly chosen $j \in \{2, 3, 4\}$, opening door $d_j$ results in a reward of 2 and for $i \notin \{1, j\}$, opening door $d_i$ results in a reward of $-2$. The agent must first choose a door after which, if it has chosen door 1, it must enter the combination (receiving a reward of 0 if it enters the incorrect combination) and, otherwise, the agent immediately receives its reward. See Fig. 1.

### A.5.2 2D-Lighthouse (2D-LH)

2D variant of the exemplar grid-world task introduced in Ex. 2, aimed to empirically verify our analysis of the imitation gap. A reward awaits at a randomly chosen corner of a square grid of size $2N + 1$ and the agent can only see the local region, a square of size $2i + 1$ about itself (an $f^i$-restricted observation). Additionally, all $f^i$ allow the agent access to it's previous action. As explained in Ex. 2, we experiment with optimizing $f^i$-policies when given supervision from $f^j$-optimal teachers (*i.e.*, experts that are optimal when restricted to $f^j$-restricted observations). See Fig. A.2 for an illustration.

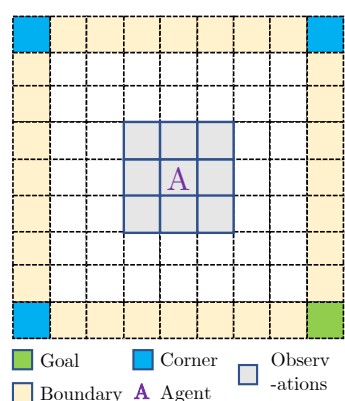

Goal · Corner · Observ
Boundary · A Agent · -ations

Figure A.2: 2D-LIGHTHOUSE

### A.5.3 LavaCrossing (LC)

Initialized on the top-left corner the agent must navigate to the bottom-right goal location. There exists at least one path from start to end, navigating through obstacles. Refer to Fig. 4 where, for illustration, we show a simpler grid. Here the episode terminates if the agent steps on any of the `lava` obstacles. This LC environment has size $25 \times 25$ with 10 `lava` rivers ('S25, N10' as per the notation of [9]), which are placed vertically or horizontally across the grid. The expert is a shortest path agent with access to the entire environment's connectivity graph and is implemented via the `networkx` python library.

### A.5.4 WallCrossing (WC)

Similar to LAVACROSSING in structure and expert, except that obstacles are `walls` instead of `lava`. Unlike `lava` (which immediately kills the agent upon touching), the agent may run into `walls` without consequence (other than wasting time). Our environment is of size $25 \times 25$ with 10 `walls` ('S25, N10').

### A.5.5 WC/LC Switch

In this task the agent faces a more challenging filtration function. In addition to navigational actions, agents for this task have a 'switch' action. Using this switch action, the agents can switch-on the lights of an otherwise darkened environment which is implemented as an observation tensor of all zeros. In WC, even in the dark, an agent can reach the target by taking random actions with non-negligible probability. Achieving this in LC is nearly impossible as random actions will, with high probability, result in stepping into `lava` and thereby immediately end the episode.

We experiment with two variants of this 'switch' – ONCE and FAULTY. In the ONCE SWITCH variant, once the the 'switch' action is taken, the lights remain on for the remainder of the episode. This is implemented as the unaffected observation tensor being available to the agent. In contrast, in the FAULTY SWITCH variant, taking the 'switch' action will only turn the lights on for a single timestep. This is implemented as observations being available for one timestep followed by zero tensors (unless the 'switch' action is executed again).

The expert for these tasks is the same as for WC and LC. Namely, the expert always takes actions along the shortest path from the agents current position to the goal and is unaffected by whether the light is on or off. For the expert-policy-based methods this translates to the learner agent getting perfect (navigational) supervision while struggling in the dark, with no cue for trying the switch

action. For the expert-demonstrations-based methods this translates to the demonstrations being populated with blacked-out observations paired with perfect actions: such actions are, of course, difficult to imitate. As FAULTY is more difficult than ONCE (and LC more difficult than WC) we set grid sizes to reduce the difference in difficulty between tasks. In particular, we choose to set WC ONCE SWITCH on a (S25, N10) grid and the LC ONCE SWITCH on a (S15, N7) grid. Moreover, WC FAULTY SWITCH is set with a (S15, N7) grid and LC FAULTY SWITCH with a (S9, N4) grid.

### A.5.6 WC/LC Corrupt

In the SWITCH task, we study agents with observations affected by a challenging filtration function. In this task we experiment with corrupting the expert's actions. The expert policy flips over to a random policy when the expert is $N_C$ steps away from the goal. For the expert-policy-based method this translates to the expert outputting uniformly random actions once it is within $N_C$ steps from the target. For the expert-demonstrations-based methods this translates to the demonstrations consisting of some valid (observation, expert action) tuples, while the tuples close to the target have the expert action sampled from a uniform distribution over the action space. WC CORRUPT is a (S25, N10) grid with $N_C = 15$, while the LC CORRUPT is significantly harder, hence is a (S15, N7) grid with $N_C = 10$.

### A.5.7 PointGoal Navigation

In PointGoal Navigation, a randomly spawned agent must navigate to a goal specified by a relative-displacement vector. The observation space is composed of rich egocentric RGB observations ($256{\times}256{\times}3$) with a limited field of view. The action space is {move_ahead, rotate_right, rotate_left, stop}. The task was formulated by [1] and implemented for the AIHABITAT simulator by [54]. Our reward structure, train/val/test splits, PointNav dataset, and implementation follow [54]. RL agents are trained using PPO following authors' implementation[7]. The IL agent is trained with on-policy behavior cloning using the shortest-path action. A static combination of the PPO and BC losses (*i.e.* a simple sum of the PPO loss and IL cross entropy loss) is also used a competing baseline for ADVISOR. Note that the agent observes a filtered egocentric observation while the shortest-path action is inferred from the entire environment state leading to a significant imitation gap. We train on the standard Gibson set of 76 scenes, and report metrics as an average over the val. set consisting of 14 unseen scenes in AIHABITAT. We use a budget of 50 million frames, *i.e.*, ∼2 days of training on 4 NVIDIA TitanX GPUs, and 28 CPUs for each method.

### A.5.8 ObjectGoal Navigation

In ObjectGoal Navigation within the RoboTHOR environment, a randomly spawned agent must navigate to a goal specified by an object category. In particular, the agent must search it's environment to find an object of the given category and take a stop action (which ends the episode regardless of success) when that object is within 1m of the agent and visible. The observation space is composed of rich egocentric RGB observations ($300{\times}400{\times}3$) with a limited field of view. The action space is {move_ahead, rotate_right, rotate_left, look_up, ,look_down, stop}. The OBJECTNAV task within the RoboTHOR environment was proposed by [14], we use the version of this task corresponding to the 2021 RoboTHOR ObjectNav Challenge[8] and use this challenge's reward structure, dataset, train/val/test splits, and their baseline model architecture. This challenge provides implementations of PPO and DAgger where the DAgger agent is trained with supervision coming from a shortest-path expert. We implement our ADVISOR methodology (with no teacher forcing) as well as a baseline where we simply sum PPO and IL losses. We use a budget of 100 million frames, *i.e.*, ∼2-5 days of training, 8 NVIDIA TitanX GPUs, and 56 CPUs for each method. At every update step we use 60 rollouts of length 128 and perform 4 gradient steps with the rollout.

### A.5.9 Cooperative Navigation

In Cooperative Navigation, there are three agents and three landmarks. The goal of the three agents is to cover the three landmarks. Agents are encouraged to move toward uncovered landmarks and get penalized when they collide with each other. Agents have limited visibility range. The agents can

---

[7]https://github.com/facebookresearch/habitat-lab
[8]https://ai2thor.allenai.org/robothor/cvpr-2021-challenge

only observe other agents and landmarks within its visibility range (euclidean distance to the agent). The action space has five dimensions. The first dimension is no-op, and the other four dimensions represent the forward, backward, left, and right force applied to the agent. The RL agents are trained with MADDPG [37] with a permutation invariant critic [36]. The IL agents are trained using DAgger. The experts are pre-trained RL agents with no limits to their visibility range. Following [37, 36], we use a budge of $1.5$ million environment steps. We use one NVIDIA GTX1080 and 2 CPUs to train these agents.

### A.5.10  Observation spaces

**2D-LH.** Within our 2D-LH environment we wish to train our agent in the context of Proposition 1 so that the agent may learn any $f$-restricted policy. As the 2D-LH environment is quite simple, we are able to uniquely encode the state observed by an agent using a $4^4 \cdot 5^2 = 6400$ dimensional $\{0, 1\}$-valued vector such that any $f$-restricted policy can be represented as a linear function applied to this observation (followed by a soft-max).[9]

**PD.** Within the PD environment the agent's observed state is very simple: at every timestep the agent observes an element of $\{0, 1, 2, 3\}$ with 0 denoting that no door has yet been chosen, 1 denoting that the agent has chosen door $d_1$ but has not begun entering the code, 2 indicating that the agent has chosen door $d_1$ and has started entering the code, and 3 representing the final terminal state after a door has been opened or combination incorrectly entered.

**MINIGRID.** The MINIGRID environments [9] enable agents with an egocentric "visual" observation which, in practice, is an integer tensor of shape $7 \times 7 \times 3$, where the channels contain integer labels corresponding to the cell's type, color, and state. Kindly see [9, 8] for details. For the above tasks, the cell types belong to the set of (`empty, lava, wall, goal`).

**POINTNAV.** Agents in the POINTNAV task observe, at every step, egocentric RGB observations ($256 \times 256 \times 3$) of their environment along with a relative displacement vector towards the goal (*i.e.* a 2d vector specifying the location of the goal relative the goal). See Figure 4 for an example of one such egocentric RGB image.

**OBJECTNAV.** Agents in the OBJECTNAV task observe, at every step, egocentric RGB observations ($300 \times 400 \times 3$) of their environment along with an object category (*e.g.* "BaseballBat") specifying their goal. See Figure 4 for an example of one such egocentric RGB image. Note that agents in the OBJECTNAV task are generally also allowed access to egocentric depth frames, we do not use these depth frames in our experiments as their use slows simulation speed.

COOPNAV. At each step, each agent in COOPNAV task observes a 14-dimensional vector, which contains the absolute location and speed of itself, the relative locations to the three landmarks, and the relative location to other two agents.

### A.6  ADVISOR can outperform BC in the no-imitation-gap setting

Recall the setting of our 2D-LH experiments in Section 4.4 where we train $f^i$-restricted policies (*i.e.*, an agent that can see $i$ grid locations away) using $f^j$-optimal teachers. In particular, we train 25 policies on each $i, j$ pair where for $1 \le i \le j \le 15$ and $i, j$ are both odd. Each trained policy is then evaluated on 200 random episodes and we record average performance across various metrics across these episodes. Complementing Fig. 6 from the main paper, Fig. A.3 shows the box plots of the trained policies average episode lengths, lower being better, when training with BC, BC$\rightarrow$ PPO, ADVISOR, and PPO (PPO does not use expert supervision so we simply report the performance of PPO trained $f^i$-restricted policies for each $i$).

As might be expected: ADVISOR has consistently low episode lengths across all $i, j$ pairs suggesting that ADVISOR is able to mitigate the impact of the imitation gap. One question that is not well-answered by Fig. A.3 is that of the relative performance of ADVISOR and IL methods when *there is no imitation gap*, namely the $i = j$ case. As ADVISOR requires the training of an auxiliary policy in addition (but, in parallel) to a main policy, we test the sample efficiency of ADVISOR head-on with IL methods. Table A.1 records the percentage of runs in which ADV, BC, and † attain near optimal (within 5%) performance when trained in the no-imitation-gap setting (*i.e.* $i = j$) for

---

[9]As the softmax function prevents us from learning a truly deterministic policy we can only learn a policy arbitrarily close to such policies. In our setting, this distinction is irrelevant.

| Method | % converged to near optimal performance | | | | | | | |
|---|---|---|---|---|---|---|---|---|
| | $i = 1$ | 3 | 5 | 7 | 9 | 11 | 13 | 15 |
| ADV | 1 | 1 | 1 | 1 | 1 | 1 | 1 | 1 |
| BC | 1 | 0.72 | 0.52 | 0.72 | 0.68 | 0.84 | 0.96 | 1 |
| † | 0.88 | 0.56 | 0.24 | 0.08 | 0.52 | 0.96 | 1 | 1 |

Table A.1: **Comparing efficiency of IL *vs*. ADVISOR in 2D-LH**. Here we report the percentage of runs (of 25 runs per (method, $i$) pair) that various methods converged to near-optimal performance (within 5% of optimal) with a budget of 300,000 training steps. Here $i$ corresponds to an $f^i$-restricted (student) policy trained with expert supervision from an $f^i$-optimal teacher (*i.e.* the 'no-imitation-gap' setting).

different grid visibility $i$. We find that only ADVISOR consistently reaches near-optimal performance within the budget of 300,000 training steps. We suspect that this is due to the RL loss encouraging early exploration that results in the agent more frequently entering states where imitation learning is easier. This interpretation is supported by the observation that ADV, BC, and † all consistently reach near-optimal performance when $i$ is very small (almost all states look identical so exploration can be of little help) and when $i$ is quite large (the agent can see nearly the whole environment so there is no need to explore). While we do no expect this trend to hold in all cases, indeed there are likely many cases where pure-IL is more effective than ADV in the no-imitation-gap setting, it is encouraging to see that ADV can bring benefits even when there is no imitation gap.

## A.7 Additional baseline details

### A.7.1 Baselines details for 2D-LH, PD, and MINIGRID tasks

In Tab. A.2, we include details about the baselines considered in this work, including – purely RL (1), purely IL ($2 - 4, 9$), a sequential combination of IL/RL ($6 - 8$), static combinations of IL/RL ($5, 10$), a method that uses expert demonstrations to generate rewards for reward-based RL (*i.e.* GAIL, 11), and our dynamic combinations ($12 - 15$). Our imitation learning baselines include those which learn from both expert policy (*i.e.* an expert action is assumed available for any state) and expert demonstrations (offline dataset of pre-collected trajectories).

In our study of hyperparameter robustness (using the PD and MINIGRID tasks) the hyperparameters (hps) we consider for optimization have been chosen as those which, in preliminary experiments, had a substantial impact on model performance. This includes the learning rate (lr), portion of the training steps devoted to the first stage in methods with two stages (stage-split), and the temperature parameter in the ADVISOR weight function ($\alpha$).[10] Note that, the random environment seed also acts as an implicit hyperparameter. We sample hyperparameters uniformly at random from various sets. In particular, we sample lr from $[10^{-4}, 0.5)$ on a log-scale, stage-split from $[0.1, 0.9)$, and $\alpha$ from $\{4, 8, 16, 32\}$.

In the below we give additional detailis regarding the GAIL and ADV$^{\text{demo}}$ + PPO methods.

**Generative adversarial imitation learning (GAIL).** For a comprehensive overview of GAIL, please see [24]. Our implementation closely follows that of Ilya Kostrikov [32]. We found GAIL to be quite unstable without adopting several critical implementation details. In particular, we found it critical to (1) normalize rewards using a (momentum-based) running average of the standard deviation of past returns and (2) provide an extensive "warmup" period in which the discriminator network is pretrained. Because of the necessity of this "warmup period", our GAIL baseline observes more expert supervision and is given a budget of substantially more gradient steps than all other methods. Because of this, our comparison against GAIL *disadvantages our ADVISOR method*. Despite this disadvantage, ADVISOR still outperforms.

**The ADV$^{\text{demo}}$ + PPO method.** As described in the main paper, the ADV$^{\text{demo}}$ + PPO method attempts to bring the benefits of our ADVISOR methodology to the setting where expert demonstrations are

---

[10]See Sec. 3.2 for definition of the weight function for ADVISOR.

[11]While implemented with supervision from expert policy, due to the teacher forcing being set to 1.0, this method can never explore beyond states (and supervision) in expert demonstrations.

| # | Method | IL/RL | Expert supervision | Hps. searched |
|---|--------|-------|--------------------|---------------|
| 1 | PPO | RL | Policy | lr |
| 2 | BC | IL | Policy | lr |
| 3 | † | IL | Policy | lr, stage-split |
| 4 | $BC^{tf=1}$ | IL | Policy[11] | lr |
| 5 | $BC + PPO$ | IL&RL | Policy | lr |
| 6 | $BC \rightarrow PPO$ | IL→RL | Policy | lr, stage-split |
| 7 | $\dagger \rightarrow PPO$ | IL→RL | Policy | lr, stage-split |
| 8 | $BC^{tf=1} \rightarrow PPO$ | IL→RL | Policy | lr, stage-split |
| 9 | $BC^{demo}$ | IL | Demonstrations | lr |
| 10 | $BC^{demo} + PPO$ | IL&RL | Demonstrations | lr |
| 11 | GAIL | IL&RL | Demonstrations | lr |
| 12 | ADV | IL&RL | Policy | lr, $\alpha$ |
| 13 | $\dagger \rightarrow ADV$ | IL&RL | Policy | lr, $\alpha$, stage-split |
| 14 | $BC^{tf=1} \rightarrow ADV$ | IL&RL | Policy | lr, $\alpha$, stage-split |
| 15 | $BC^{demo} + ADV$ | IL&RL | Demonstrations | lr, $\alpha$ |

Table A.2: **Baseline details.** IL/RL: Nature of learning, Expert supervision: the type of expert supervision leveraged by each method, Hps. searched: hps. that were randomly searched over, fairly done with the same budget (see Sec. A.9 for details).

available but an expert policy (*i.e.*, an expert that can be evaluated at arbitrary states) is not. Attempting to compute the ADVISOR loss (recall Eq. (2)) on off-policy demonstrations is complicated however, as our RL loss assumes access to on-policy demonstrations. In theory, importance sampling methods, see, *e.g.*, [39], can be used to "reinterpret" expert demonstrations as though they were on-policy. But such methods are known to be somewhat unstable, non-trivial to implement, and may require information about the expert policy that we do not have access to. For these reasons, we choose to use a simple solution: when computing the ADVISOR loss on expert demonstrations we ignore the RL loss. Thus $ADV^{demo} + PPO$ works by looping between two phases:

- Collect an (on-policy) rollout using the agent's policy, compute the PPO loss for this rollout and perform gradient descent on this loss to update the parameters.
- Sample a rollout from the expert demonstrations and, using this rollout, compute the demonstration-based ADVISOR loss

$$\mathcal{L}^{ADV\text{-}demo}(\theta) = \mathbb{E}_{demos.}[w(S) \cdot CE(\pi^{teach}(S), \pi_f(S;\theta))], \qquad (10)$$

and perform gradient descent on this loss to update the parameters.

### A.7.2 Baselines used in POINTNAV experiments

Our POINTNAV baselines are described in Appendix A.5.9. See also Table A.4.

### A.7.3 Baselines details for OBJECTNAV experiments

Our OBJECTNAV baselines are described in Appendix A.5.8. See also Table A.4.

### A.7.4 Baselines used in COOPNAV experiments

Our COOPNAV baselines are described in Appendix A.5.9. We follow the implementation of [36].

### A.8 Architecture Details

**2D-LH model.** As discussed in Sec. A.5.10, we have designed the observation given to our agent so that a simple linear layer followed by a soft-max function is sufficient to capture any $f$-restricted policy. As such, our main and auxiliary actor models for this task are simply linear functions mapping the input 6400-dimensional observation to a 4-dimensional output vector followed by a soft-max non-linearity. The critic is computed similarly but with a 1-dimensional output and no non-linearity.

**PD model.** Our PD model has three sequential components. The first embedding layer maps a given observation, a value in $\{0, 1, 2, 3\}$, to an 128-dimensional embedding. This 128-dimensional vector is then fed into a 1-layer LSTM (with a 128-dimensional hidden state) to produce an 128-output representation $h$. We then compute our main actor policy by applying a $128 \times 7$ linear layer followed by a soft-max non-linearity. The auxiliary actor is produced similarly but with separate parameters in its linear layer. Finally the critic's value is generated by applying a $128 \times 1$ linear layer to $h$.

**MINIGRID model.** Here we detail each component of the model architecture illustrated in Fig. 3. The encoder ('Enc.') converts observation tensors (integer tensor of shape $7 \times 7 \times 3$) to a corresponding embedding tensor via three embedding sets (of length 8) corresponding to type, color, and state of the object. The observation tensor, which represents the 'lights-out' condition, has a unique (*i.e.*, different from the ones listed by [9]) type, color and state. This prevents any type, color or state from having more than one connotation. The output of the encoder is of size $7 \times 7 \times 24$. This tensor is flattened and fed into a (single-layered) LSTM with a 128-dimensional hidden space. The output of the LSTM is fed to the main actor, auxiliary actor, and the critic. All of these are single linear layers with output size of $|\mathcal{A}|$, $|\mathcal{A}|$ and 1, respectively (main and auxiliary actors are followed by soft-max non-linearities).

**POINTNAV, OBJECTNAV, and COOPNAV model.**

For the POINTNAV [54], OBJECTNAV [14], and COOPNAV [36] tasks, we (for fair comparison) use model architectures from prior work. For use with ADVISOR, these model architectures require an additional auxiliary policy head. We define this auxiliary policy head as a linear layer applied to the model's final hidden representation followed by a softmax non-linearity.

## A.9 Fair Hyperparameter Tuning

As discussed in the main paper, we attempt to ensuring that comparisons to baselines are fair. In particular, we hope to avoid introducing misleading bias in our results by extensively tuning the hyperparameters (hps) of our ADVISOR methodology while leaving other methods relatively un-tuned.

**2D-LH: Tune by Tuning a Competing Method.** The goal of our experiments with the 2D-LH environment are, principally, to highlight that increasing the imitation gap can have a substantial detrimental impact on the quality of policies learned by training IL. Because of this, we wish to give IL the greatest opportunity to succeed and thus we are not, as in our PD/MINIGRID experiments, attempting to understand its expected performance when we must search for good hyperparameters. To this end, we perform the following procedure for every $i, j \in \{1, 3, 5 \ldots, 15\}$ with $i < j$.

For every learning rate $\lambda \in \{100$ values evenly spaced in $[10^{-4}, 1]$ on a log-scale$\}$ we train a $f^i$-restricted policy to imitate a $f^j$-optimal teacher using BC. For each such trained policy, we roll out trajectories from the policy across 200 randomly sampled episodes (in the 2D-LH there is no distinction between training, validation, and test episodes as there are only four unique initial world settings). For each rollout, we compute the average cross entropy between the learned policy and the expert's policy at every step. A "best" learning rate $\lambda^{i,j}$ is then chosen by selecting the learning rate resulting in the smallest cross entropy (after having smoothed the results with a locally-linear regression model [70]).

A final learning rate is then chosen as the average of the $\lambda^{i,j}$ and this learning rate is then used when training all methods to produce the plots in Fig. 6. As some baselines require additional hyperparameter choices, these other hyperparameters were chosen heuristically (post-hoc experiments suggest that results for the other methods are fairly robust to these other hyperparameters).

**PD and MINIGRID tasks: Random Hyperparameter Evaluations.** As described in the main paper, we follow the best practices suggested by Dodge et al. [15]. In particular, for our PD and MINIGRID tasks we train each of our baselines when sampling that method's hyperparameters, see Table A.2 and recall Sec. A.7, at random 50 times. Our plots, *e.g.*, Fig. 5, then report an estimate of the expected (validation set) performance of each of our methods when choosing the best performing model from a fixed number of random hyperparameter evaluations. Unlike [15], we compute this estimate using a U-statistic [64, Chapter 12] which is unbiased. Shaded regions encapsulate the 25-to-75th quantiles of the bootstrap distribution of this statistic.

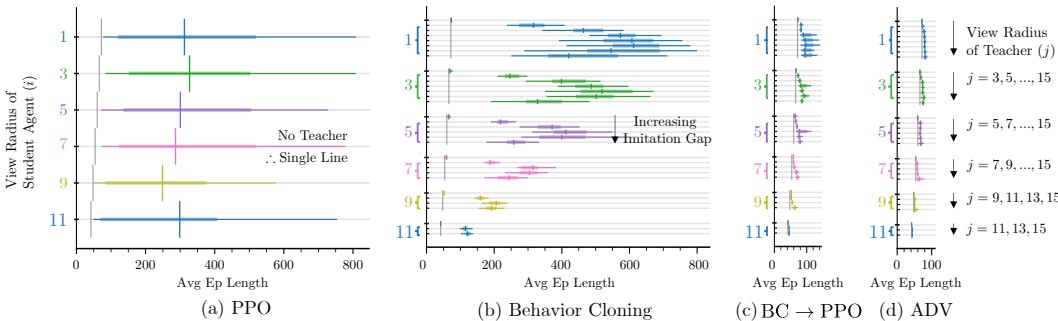

Figure A.3: **Size of the imitation gap directly impacts performance.** Training $f^i$-restricted students with $f^j$-optimal teachers (in 2D-LH).

POINTNAV, OBJECTNAV, **and** COOPNAV **tasks: use hyperparameters from in prior work.** Due to computational constraints, our strategy for choosing hyperparameters for the POINTNAV, OBJECTNAV, and COOPNAV tasks was simply to follow prior work whenever possible. Of course, there was no prior work suggesting good hyperparameter values for the $\alpha, \beta$ parameters in our new ADVISOR loss. Following the intuitions we gained from our the 2D-LH, PD, and MINIGRID experiments, we fixed $\alpha, \beta$ to (10, 0.1) for POINTNAV, $\alpha, \beta$ to (20, 0.1) for OBJECTNAV, and $\alpha, \beta$ to (0.01, 0) for COOPNAV. For the OBJECTNAV task, we experimented with setting $\beta = 0$ and found that the change had essentially no impact on performance (validation-set SPL after $\approx$ 100Mn training steps actually improved slightly from .1482 to .1499 when setting $\beta = 0$).

## A.10 Training Implementation

As discussed previously, for our POINTNAV, OBJECTNAV, and COOPNAV experiments, we have used standard training implementation details (*e.g.* reward structure) from prior work. Thus, in the below, we provide additional details only for the 2D-LH, PD, and MINIGRID tasks.

A summary of the training hyperparameters and their values is included in Tab. A.3. Kindly see [58] for details on PPO and [57] for details on generalized advantage estimation (GAE).

**Max. steps per episode.** The maximum number of steps allowed in the 2D-LH task is 1000. Within the PD task, an agent can never take more than 11 steps in a single episode (1 action to select the door and then, at most, 10 more actions to input the combination if $d_1$ was selected) and thus we do not need to set a maximum number of allowed steps. The maximum steps allowed for an episode of WC/LC is set by [9, 8] to $4S^2$, where $S$ is the grid size. We share the same limits for the challenging variants – SWITCH and CORRUPT. Details of task variants, their grid size, and number of obstacles are included in Sec. A.5.

**Reward structure.** Within the 2D-LH task, the agent receives one of three possible rewards after every step: when the agent finds the goal it receives a reward of 0.99, if it otherwise has reached the maximum number of steps (1000) it receives a $-1$ reward, and otherwise, if neither of the prior cases hold, it obtains a reward of $-0.01$. See Sec. A.5.1 for a description of rewards for the PD task. For WC/LC, [9, 8] configure the environment to give a 0 reward unless the goal is reached. If the goal is reached, the reward is $1 - \frac{\text{episode length}}{\text{maximum steps}}$. We adopt the same reward structure for our SWITCH and CORRUPT variants as well.

**Computing infrastructure.** As mentioned in Sec. 4.3, for all tasks (except LH) we train 50 models (with randomly sampled hps) for each baseline. This amounts to 750 models per task or 6700 models in total. For each task, we utilize a `g4dn.12xlarge` instance on AWS consisting of 4 NVIDIA T4 GPUs and 48 CPUs. We run through a queue of 750 models using $\approx$ 40 processes. For tasks set in the MINIGRID environments, models each require $\approx 1.2$ GB GPU memory and all training completes in 18 to 36 hours. For the PD task, model memory footprints are smaller and training all models is significantly faster ($<$ 8 hours).

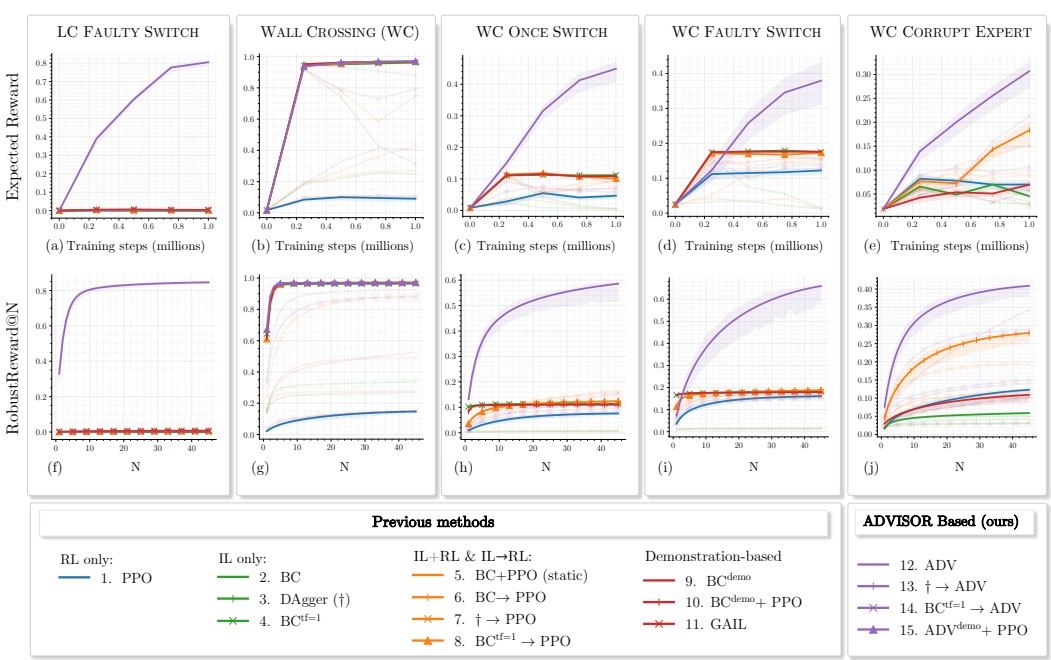

Figure A.4: **Additional results for MINIGRID tasks.** Here we include the results on the MINIGRID tasks missing from Figure 5.

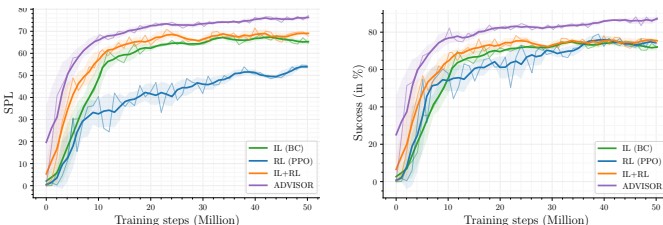

Figure A.5: **Learning curves for POINTNAV.** SPL (scaled by 100) and success rate (in %) are plotted *vs.* training steps, following the standard protocols. We evaluate checkpoints after every 1024k frames of experience. This is plotted as the thin line. The thick line and shading depicts the rolling mean (with a window size of 2) and corresponding standard deviation.

## A.11 Additional results

Here we record additional results that were summarized or deferred in Section 4.4. In particular,

- Figure A.3 complements Figure 6 from the main paper and provides results for additional baselines on the 2D-LH task. Notice that both the pipelined IL→PPO and ADVISOR methods greatly reduce the impact of the imitation gap (Figures A.3c and A.3d versus Figure A.3b) but our ADVISOR method is considerably more effective in doing so (Figure A.3c v.s. Figure A.3d).

- Figure A.4 shows the results on our remaining MINIGRID tasks missing from Figure 5. Notice that the trends here are very similar to those from Figure 5, ADVISOR-based methods have similar or better performance than our other baselines.

- Table A.5 shows an extended version of Table 1 where, rather than grouping methods together, we display results for each method individually.

- Figure A.5 displays validation set performance of our POINTNAV baselines over training. Note that static combination of RL and IL losses improves individual RL/IL baselines. Our adaptive combination of these losses (ADVISOR) outperforms these baselines and is more sample efficient.

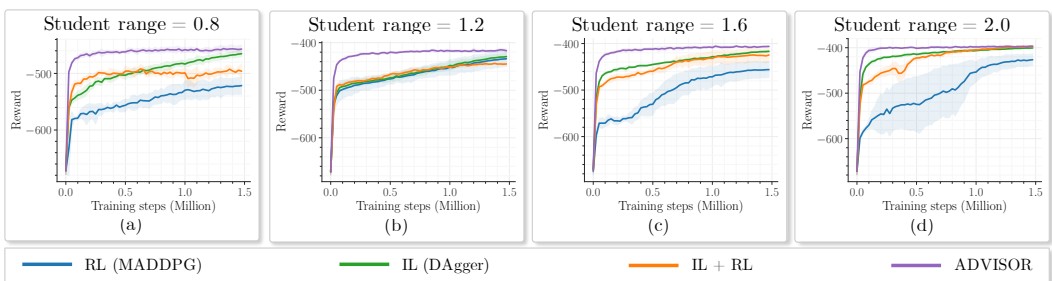

Figure A.6: **Learning curves for** COOPNAV. Rewards are plotted *vs*. training steps, following the standard protocols. For a full-range teacher, we train students with different (and limited) visibility range of 0.8, 1.2, 1.8, and 2.0. The networks are initialized with four different seeds and the mean and standard deviation are plotted. Checkpoints are evaluated at every 25k steps.

| Hyperparamter | Value |
|---|---|
| *Structural* | |
| Cell type embedding length | 8 |
| Cell color embedding length | 8 |
| Cell state embedding length | 8 |
| RNN type | LSTM |
| RNN layers | 1 |
| RNN hidden size | 128 |
| # Layers in critic | 1 |
| # Layers in actor | 1 |
| *PPO* | |
| Clip parameter ($\epsilon$) [58] | 0.1 |
| Decay on $\epsilon$ | Linear$(1, 0)$ |
| Value loss coefficient | 0.5 |
| Discount factor ($\gamma$) | 0.99 |
| GAE parameter ($\lambda$) | 1.0 |
| *Training* | |
| Rollout timesteps | 100 |
| Rollouts per batch | 10 |
| # processes sampling rollouts | 20 |
| Epochs | 4 |
| Optimizer | Adam [29] |
| ($\beta_1, \beta_2$) for Adam | (0.9, 0.999) |
| Learning rate | searched |
| Gradient clip norm | 0.5 |
| Training steps (WC/LC & variants) | $1 \cdot 10^6$ |
| Training steps (2D-LH & PD) | $3 \cdot 10^5$ |

Table A.3: Structural and training hyperparameters for 2D-LH, PD, and MINIGRID tasks.

- Figure A.6 lays out the performance of agents on the COOPNAV task. In the main paper we include results for the limited visibility range of 1.6 for the student. Here, we include results for four visibility range. RL only baseline is least sample-efficient. Overall, we find ADVISOR is significantly more sample efficient – most of the learning is completed in just 0.2 million steps while the other baselines take over 1.5 million steps.

| Hyperparamter | POINTNAV | OBJECTNAV |
|---|---|---|
| *Structural* | | |
| RNN type | GRU | |
| RNN layers | 1 | |
| RNN hidden size | 512 | |
| # Layers in critic | 1 | |
| # Layers in actor | 1 | |
| *PPO* | | |
| Clip parameter ($\epsilon$) [58] | 0.1 | |
| Decay on $\epsilon$ | None | |
| Value loss coefficient | 0.5 | |
| Discount factor ($\gamma$) | 0.99 | |
| GAE parameter ($\lambda$) | 0.95 | |
| *Training* | | |
| Rollout timesteps | 128 | |
| Rollouts per batch | 60 | 8 |
| # processes sampling rollouts | 60 | 16 |
| Epochs | 4 | |
| Optimizer | Adam [29] | |
| ($\beta_1$, $\beta_2$) for Adam | (0.9, 0.999) | |
| Learning rate | $3 \cdot 10^{-4}$ | $2.5 \cdot 10^{-4}$ |
| Gradient clip norm | 0.5 | 0.1 |
| Training steps | $100 \cdot 10^6$ | $50 \cdot 10^6$ |

Table A.4: Structural and training hyperparameters for POINTNAV and OBJECTNAV.

| Tasks → | PD | LAVACROSSING | | | | WALLCROSSING | | | |
|---|---|---|---|---|---|---|---|---|---|
| Training routines ↓ | - | Base Ver. | Corrupt Exp. | Faulty Switch | Once Switch | Base Ver. | Corrupt Exp. | Faulty Switch | Once Switch |
| PPO | 0 | 0 | 0 | 0.01 | 0 | 0.09 | 0.07 | 0.12 | 0.05 |
| BC | -0.6 | 0.1 | 0.02 | 0 | 0 | 0.25 | 0.05 | 0.01 | 0.01 |
| DAgger (†) | -0.59 | 0.14 | 0.02 | 0 | 0 | 0.31 | 0.03 | 0.01 | 0.01 |
| BC$^{\text{tf}=1}$ | -0.62 | 0.88 | 0.02 | 0.02 | 0 | 0.96 | 0.03 | 0.17 | 0.11 |
| BC+PPO (static) | -0.59 | 0.12 | 0.08 | 0 | 0 | 0.27 | 0.09 | 0.01 | 0 |
| BC→ PPO | -0.17 | 0.15 | 0.32 | 0.02 | 0 | 0.43 | 0.18 | 0.14 | 0.09 |
| † → PPO | -0.45 | 0.32 | 0.61 | 0.02 | 0 | 0.75 | 0.15 | 0.15 | 0.1 |
| BC$^{\text{tf}=1}$ → PPO | -0.5 | 0.94 | 0.74 | 0.04 | 0 | **0.97** | 0.09 | 0.17 | 0.1 |
| BC$^{\text{demo}}$ | -0.62 | 0.88 | 0.02 | 0.02 | 0 | 0.96 | 0.07 | 0.18 | 0.11 |
| BC$^{\text{demo}}$+ PPO | -0.64 | **0.96** | 0.2 | 0.02 | 0 | **0.97** | 0.03 | 0.17 | 0.11 |
| GAIL | -0.09 | 0 | 0 | 0.02 | 0 | 0.11 | 0.06 | 0.16 | 0.07 |
| ADV | **1** | 0.18 | 0.8 | **0.77** | **0.8** | 0.41 | **0.31** | **0.38** | **0.45** |
| BC$^{\text{tf}=1}$ → ADV | -0.13 | 0.55 | 0.83 | 0.02 | 0 | 0.88 | 0.15 | 0.15 | 0.09 |
| † → ADV | -0.1 | 0.47 | 0.73 | 0.01 | 0 | 0.79 | 0.21 | 0.13 | 0.07 |
| ADV$^{\text{demo}}$+ PPO | 0 | **0.96** | **0.94** | 0.03 | 0 | **0.97** | 0.11 | 0.14 | 0.06 |

Table A.5: **Expected rewards for the POISONEDDOORS task and MINIGRID tasks.** Here we show an expanded version of Table A.5 where results for all methods rather than grouped methods. For each of our 15 training routines we report the expected maximum validation set performance (when given a budget of 10 random hyperparameter evaluations) after training for ≈300k steps in the POISONEDDOORS environment and ≈1Mn steps in our 8 MINIGRID tasks. The maximum possible reward is 1 for the MINIGRID tasks.