# OpenReview forum: "Bridging the Imitation Gap by Adaptive Insubordination"
_NeurIPS.cc/2021/Conference — NeurIPS 2021 Poster_

### Official Review · Reviewer_Hfmt · 2021-07-05

**Rating:** 6
**Confidence:** 4

**Summary:**

This paper points out the existence of "imitation gap" when a teacher and student policy has different observations. Due to this imitation gap, the student policy cannot determine which action leads to a higher reward, and thus ends up outputting a uniformly random action or average actions of the teacher policy.

The proposed method, ADVISOR, tackles such imitation gap by balancing between learning from reward and expert. When the limited observation is enough to predict the optimal action, the student policy imitates the demonstrations; otherwise, it learns from reward. As the weight function, ADVISOR uses the divergence between the teacher policy and the auxiliary policy, which solely learns from the demonstrations.

**Limitations And Societal Impact:**

Adequately addressed.

**Main Review:**

**Originality**: This paper identifies a new concept of "imitation gap", which can easily happen and degrade imitation learning performance, and proposes a simple method to get around the imitation gap.

**Clarity**: The paper is clearly written with intuitive examples.

**Quality**: The exhaustive experiments on diverse environments demonstrate the existence of the imitation gap and ADVISOR's superior performance under the imitation gap.

**Significance**: Overall, the paper proves the importance of dealing with "imitation gap" and the proposed method effectively learns toy tasks under the imitation gap. This tells us that when we design an expert policy or collect expert demonstrations, we must be aware of the effect of such imitation gaps and try to minimize the gaps. However, it is not very clear how important this imitation gap is in complex environments, such as PointGoal and ObjectGoal navigation tasks, as the performance gap is not significant.


Here are a few questions regarding the method and experiments.

- The only imitation gap considered in this paper is about the observational difference between the teacher and student. However, the gap between the teacher and student could be often aroused from different dynamics, embodiments, and environment stochasticity. Can this proposed method handle such gaps as well, especially under stochastic environments?
- The experiments are mostly focused on navigation. A few experiments on common continuous control benchmarks (locomotion or manipulation) could make the paper stronger.
- The key of ADVISOR is to ignore the expert policy on states which are not reproducible in the student's observation space. This highly relies on the accurate auxiliary policy. To learn a more accurate auxiliary policy, use of a replay buffer can be more appropriate. What is the rationale of using on-policy data, which can make the auxiliary policy biased toward recent trajectories?
- The weight function used in ADVISOR is somewhat complicated and requires two additional hyperparameters to be examined. Is there any benefit or explanation of the use of this complex function compared to simple linear, quadratic, or exponential ones?

### Minor comments and questions

- In some places, the white spaces between lines and figures are too compact to read comfortably.
- In Table 2, "e" is added to the second column.

**Time Spent Reviewing:**

3

---

> ### Author Response · Authors · 2021-08-10
> **Response to Reviewer Hfmt**
>
> Thank you for your thoughtful consideration of our work and feedback. Our response to your questions and comments is included below.
>
> -------
>
> > The only imitation gap considered in this paper is about the observational difference between the teacher and student. However, the gap between the teacher and student could be often aroused from different dynamics, embodiments, and environment stochasticity. Can this proposed method handle such gaps as well, especially under stochastic environments?
>
> This is an excellent observation: there are many orthogonal dimensions along which teachers and their students may differ and these each may induce a different type of "imitation gap." ADVISOR was designed with observational differences in mind and thus a naïve application of ADVISOR would be unlikely to bridge these gaps. That said, ADVISOR-like ideas seem quite generally applicable:
> * **Different dynamics and stochasticity** - rather than learning an auxiliary policy, learn an auxiliary dynamics model $T^{\text{aux}}(s, a, s')$ which predicts the probability of the teacher observing a state $s'$ after taking action $a$ in state $s$. We can then only imitate the teacher in those states where the dynamics of the student and expert are similar (using $T^{\text{aux}}(s, a, s')$ to measure this similarity).
> * **Different embodiments** - this is somewhat more challenging to describe generally (as there are myriad means by which embodiment may differ) but, for example, suppose that the teacher has an arm with a somewhat larger reach than the student. We might then learn an auxiliary "reachability" function $\text{reach}^{\text{aux}}(s, p)$ which determines if a point $p$ is reachable by the teacher. We then might only imitate the teacher when the teacher's target arm position is reachable by both the teacher and student simultaneously.
> * **Different memory mechanisms** - while we have not emphasized this in our main paper, ADVISOR does actually extend to cases where the teacher and student have different memory mechanisms, e.g. the teacher builds a map of the environment while the student is reactive.
>
> ------
>
> > The experiments are mostly focused on navigation. A few experiments on common continuous control benchmarks (locomotion or manipulation) could make the paper stronger.
>
> Our focus on navigational tasks is largely pedagogical and based on how researchers use these environments in practice: it is straightforward to see how an observation-based imitation gap can occur in a navigational setting as it is natural to envision a teacher who can "see the whole map" while the student has access only to local observations. Indeed this is precisely what is done in practice when training agents to follow the shortest paths to the goal. Within the setting of continuous control (e.g. any of the many popular benchmarks like MuJoCo (Todorov et al., 2012), dm_control suite (Tassa et al., 2020), meta-world (Yu et al., 2019), rllab/garage (Duan et al., 2016)) the imitation gap is a bit more subtle. For example, most work in MuJoCo provides the agent the full environment state in the form of joint angles and orientations. If the student has access to these standard observations then there is no imitation gap. If, on the other hand, the student must act from pixels while the teacher was trained using the full environment state then an imitation gap will exist. We expect that ADVISOR would help in this setting and leave this direction of exploration for future work.
>
> While we believe that our existing experiments are quite convincing and we have provided evidence that ADVISOR can work in continuous control setting (recall that our experiments on Cooperative Navigation, Tab. 2 and Fig. 4 are continuous), we do agree that an exploration of ADVISOR in the continuous space would be quite interesting. Our code is fully compatible with the OpenAI Gym (Brockman et al., 2016) which should make it easy for researchers to adopt ADVISOR for their own experiments.
>
> ------
>
> > The key of ADVISOR is to ignore the expert policy on states which are not reproducible in the student's observation space. This highly relies on the accurate auxiliary policy. To learn a more accurate auxiliary policy, the use of a replay buffer can be more appropriate. What is the rationale of using on-policy data, which can make the auxiliary policy biased toward recent trajectories?
>
> ADVISOR is completely compatible with a replay-buffer and you're entirely correct that this will likely improve the auxiliary policy and thereby the training of the main policy. We chose to primarily focus on the on-policy formation for two reasons:
>
> 1. **Equalizing gradient updates across experiments** - we were concerned that allowing for  _additional_ off-policy updates with a replay-buffer would unfairly advantage ADVISOR against our other baseline methods. We will add a discussion that using a replay buffer can further improve the sample-efficiency of ADVISOR.
> 2. **Relative engineering complexity** - many replay-buffer RL implementations assume that models are reactive and, in our experience, can be a bit tricky to engineer correctly when one uses recurrent neural networks (as we do).
>
> Additional experiments with replay-buffers would further strengthen our contributions. Nevertheless, we believe that our extensive experiments in the on-policy setting provide strong evidence for the advantages of ADVISOR.
>
> ------
>
> > The weight function used in ADVISOR is somewhat complicated and requires two additional hyperparameters to be examined. Is there any benefit or explanation of the use of this complex function compared to simple linear, quadratic, or exponential ones?
>
> This constructive feedback will improve the understanding of our work.
>
> * Hyperparameters -  Our weight function, recall Eq. (2), has two hyperparameters, $\alpha$ (similar to a standard temperature parameter) and $\beta$ (used to threshold). To simplify this weight function, we have tried setting $\beta=0$ in our experiments and found that it had little to no impact on our results (note that results in our appendix, see L937-939, already suggested the marginal impact of $\beta$ for Object Navigation). As $\beta$ appears unnecessary for good performance, we will revise Sec. 3.2 to remove it and thereby simplify our weight function to a simple exponential one.
>
> * Other weight functions - We chose to use the exponential function (instead of linear, quadratic, etc. ones) largely as it "cancels out" the log term in the KL divergence which leads to a more interpretable weight in some settings. Recall Eq. (3): if (1) the teacher is discrete and deterministic, (2) $\alpha=1$, and (3) $\beta=0$, then the use of the exponential in the weight function results in the weight equalling the probability of the auxiliary policy choosing the teacher's action. In this case, varying $\alpha$ is equivalent to simply varying the temperature of this probability. We hope to explore other weight functions (and their probabilistic interpretations) in future work.
>
> -------
>
> > Minor comments and questions
>
> Thank you for this feedback. We’ll revise and incorporate this feedback.

---

> > ### Comment · Reviewer_Hfmt · 2021-08-11
> > **Response to the authors**
> >
> > Thank you for your detailed answer. The answer addressed most of my questions and I think the paper will be in a good form once including the discussion in the rebuttal. Yet, I believe the paper can have much broader impact and better usability by showing (1) experiments on manipulation where a student learns from pixels and (2) results when training an auxiliary policy using off-policy data.

---

### Official Review · Reviewer_JVEf · 2021-07-14

**Rating:** 8
**Confidence:** 4

**Summary:**

This work introduces ADVISOR, a simple yet effective approach to adaptively combining RL with supervision from imitation learning. The approach is motivated by the issue of *information gaps,* which describes the case where the policy generating the supervision has more information about the state than does the policy being trained. The authors motivate their approach as a way to address this issue and contribute a thorough set of empirical analyses to demonstrate its effectiveness. In addition, this work provides a concrete demonstration that the information gap is an obstacle.

**Limitations And Societal Impact:**

Yes, adequately addressed.

**Main Review:**

I consider this a strong paper. It clearly motivates the problem as well as the solution it proposes (which itself is attractively simple), making use of concrete demonstration throughout. It is one of the rare papers that seems to get *better* as you read it.

I lack the specific expertise to fully evaluate the originality of the work, but, as best I can tell, it should be considered novel. At the very least, the combination of contributions makes the aggregate work in the paper original.

The ideas in the paper are presented quite clearly. The authors use examples well and provide clear formalizations where useful. The paper does not rely on the appendix, but it contains plenty of clarifying information nevertheless. The main results figure (Fig 5) is a bit hard to parse.

A major strength of the paper is its experimental rigor and use of illustrative "toy" settings. The adherence to best practices makes the results more meaningful and impactful. The analysis presented in Figure 6 is particularly nice and helps to establish the significance of the information gap.

One issue with the paper is that it seems to be particularly focused on the settings where it would be possible to query the teacher policy at every state. Often, I would think, that is not the case. There is at least one experiment using ADVISOR with a fixed set of teacher demonstrations, so the method can handle this case. My sense is that it would strengthen the potential impact of the paper if deeper analysis were presented of how ADVISOR fares in these demonstration settings.

**Time Spent Reviewing:**

1.5

---

> ### Author Response · Authors · 2021-08-10
> **Response to Reviewer JVEf**
>
> We are happy to know that the reviewer found this to be a strong paper and appreciate their valuable feedback. We address their comments below:
>
> --------
>
> > The main results figure (Fig 5) is a bit hard to parse.
>
> We agree. We tried to highlight best-performing baselines as a way to improve readability while maintaining experimental rigor. We’ll improve on the readability of this figure in a revised version.
>
> --------
>
> > There is at least one experiment using ADVISOR with a fixed set of teacher demonstrations, so the method can handle this case. My sense is that it would strengthen the potential impact of the paper if deeper analysis were presented of how ADVISOR fares in these demonstration settings.
>
> While our focus in this paper was primarily on the ‘on-policy’ setting where expert supervision is available at all states, we study multiple demonstration-based baselines (including [24]) and a demonstration-based ADVISOR. We are very interested in applying the ideas from ADVISOR to off-policy demonstration-based learning more generally. As a step in this direction, in a revised version, we will include a case study where we compare the performance of $\text{ADV}^{\text{demo}}+\text{PPO}$ (against other IL methods) as we vary the size of the demonstration dataset.

---

### Official Review · Reviewer_Rksz · 2021-07-21

**Rating:** 7
**Confidence:** 4

**Summary:**

The paper approaches the problem of imitation gap. In situations where the teacher (an expert) has access to privileged information, the student may not have such comfort. Therefore, blindly imitating the teacher may lead to inefficient, unwanted or incorrect behavior of the student. The paper provides examples of such situations. Actually, sometimes the student may not even be able to imitate the teacher in some states as not all the information is available.

This leads to an imitation gap. Performing classic imitation learning in states where the imitation gap is significant can be pointless. The authors propose to try existing RL methods in such situations.

Specifically, the imitation loss and the RL loss are weighted by a normalized coefficient. One of the main contributions of the paper is how to dynamically determine the value of the coefficient. The idea is to in some sense 'measure' or 'quantify' the imitation gap in a given state. The authors follow the intuition that for states where the imitation gap is substantial, learning well the teacher's distribution over actions may not be achievable. It means the divergence between the teacher's distribution and the imitation distribution of actions will be relatively large. This quantity is used as an estimation of imitation gap.

To compute that quantity an approximation to the imitation policy is introduced. The divergence between the teacher and this approximation is used to estimate the imitation gap.


**Limitations And Societal Impact:**

The paper has a concrete chapter addressing limitations and social impact. It is clear that authors had these in their mind and addressed potential concerns.

**Main Review:**

The paper is very clear to read and understand. The flow is logical and smooth. The problem is well stated and supported by a clear example.

The approach is precisely described. The notations is consistent and easy to comprehend.

The method is practical in the sense that it takes under consideration that computing the exact imitation policy is not possible. Instead, it offers a simple and efficient way of approximating it. On the other hand it is flexible as not-critical elements of the algorithm are abstracted away. For example, there are multiple possible distance functions between the teacher policy and the imitation policy (or its approximation).

The paper introduces a novel approach to estimating the coefficient between imitation learning and reinforcement learning for states that have a significant imitation gap.

The results are on the example problems and on more challenging environments. The proposed approach performs well in the basic and more complex circumstances.

My rating is based on the proposed approach but also on the significance of the problem itself.

Of course it would be great to see the method evaluated on even more complex environments and challenges. Nevertheless, that probably will be the topic of future work. At this point the presented experiments illustrate the problem and the presented approach is able to make significant progress.

One comment is that it would be beneficial to include more analysis of the intuitions behind the imitation gap and the suggested way of quantifying it. For example, where does the quantification method perform well and what are the cases where it is imperfect? Why? What should be considered to improve over the proposed algorithm? What are the edge cases?








**Time Spent Reviewing:**

7

---

> ### Author Response · Authors · 2021-08-10
> **Response to Reviewer Rksz**
>
> Thank you for your close read of our work and your insightful feedback. We directly address your questions below:
>
> --------
>
> > include more analysis of the intuitions behind the imitation gap … where does the quantification method perform well and what are the cases where it is imperfect?
>
> We highlight two possible improvements for our method for quantifying the imitation gap. As a quick refresher, recall that, with reference to Fig. 3 and Sec. 3.3, the auxiliary policy head estimates the best $f$-restricted policy under IL (i.e. it estimates $\pi_{f}^{\text{IL}}$). We optimize the ADVISOR loss (Eq. 2). The loss has two moving parts -- the main policy $\pi_{f}$ and the auxiliary policy $\pi^{\text{aux}}_f$.
>
> 1. Potentially slow on-policy learning of the auxiliary policy
> - There are, effectively, two-phases of imitation learning in ADVISOR: first the auxiliary policy must learn to imitate the expert and, once it is able to do so, only then does the main policy begin receiving expert supervision. While this two-phase process has not seemed to result in notable sample-efficiency losses in our experiments, it has the potential to slow training (especially when the imitation gap is small). For instance, if there is no imitation gap, then having to wait for the auxiliary policy to first learn the expert policy is clearly suboptimal. We have some suggestions for future work to address this challenge (discussed below). To provide some additional quantitative analysis of this potential inefficiency we will add an additional study, within the 2D-LH environment, where we compare the difference in efficiency between pure-IL versus ADVISOR in the no-imitation-gap setting.
>
> 2. The problem of overfitting
> - Overfitting is an issue throughout machine learning and ADVISOR is, unsurprisingly, not immune: if the auxiliary policy is able to overfit to the expert on the training dataset then ADVISOR simply reduces to imitation learning. This may especially be a problem when we use our demonstrations-based ADVISOR ($\text{ADV}^{\text{demo}} + \text{PPO}$) variant with a small number of expert demonstrations. All of the usual approaches towards reducing overfitting can, of course, be applied in this setting (e.g. data augmentation) but another interesting approach may be to prevent the flow of gradients from the auxiliary loss in the shared backbone (i.e. $r_{\lambda}$, see Fig. 3). E.g. if we only allow these gradients to update the (linear) auxiliary actor head then the auxiliary policy must learn to imitate the expert using representations learned primarily through the RL loss. This should slow overfitting, perhaps allowing us to escape this local optimum.
>
> --------
>
> > What should be considered to improve over the proposed algorithm? What are the edge cases?
>
> We consider two pertinent ways to improve the proposed algorithm targeting the problem of "potentially slow on-policy learning of the auxiliary policy" described above.
>
> 1. **Better distance measures**:
> Recall, in L200-207 (Sec. 3.2), we introduced a distance $d^{0}(\pi, \pi_f)(s)$ between the teacher policy $\pi^{\text{teach}}$ and an _$f$-restricted_ policy $\pi_{f}$. We use this to estimate the distance between the auxiliary actor policy $\pi^{\text{aux}}_f$ and the teacher policy (Sec. 3.3). We make the consideration that $d^{0}(\pi, \pi_f)(s) = d(\pi(s), \pi_f(s))$ and utilize KL-divergence as $d$.
>
> * **Same penalty for the same restricted observation**:
> If two different (unfiltered) environment states $s,s'$ are mapped to the same observation by the filtration function $f$, then it is intuitive to expect that the distance between the _$f$-restricted_ policy $\pi_f$ and the teaching policy $\pi^{\text{teach}}$ should be the same at these two states. Concretely, $f(s)=f(s')$ should imply that $d^{0}(\pi^{\text{teach}},\pi_f)(s)=d^{0}(\pi^{\text{teach}},\pi_f)(s')$. While our $d^0$ is simple to estimate, it doesn’t have this property. In Appendix A.3, we indicate ways to potentially improve our distance measure by incorporating this property. Further experiments would be required to determine if this new distance measure was empirically more efficient. We will include a clearer reference to these ideas (and Appendix A.3) in the main paper, in a revised version.
>
> * **Prospective estimation of when a state is imitable**:
> As discussed previously, ADVISOR may lose sample efficiency simply because we must wait for the auxiliary policy to learn from the teacher before we are able to provide such supervision to the acting policy. In Appendix A.4 we discuss a potential direction in which we learn to predict in which states the auxiliary policy will be able to imitate the expert _at the end of training_ thus allowing us to skip having to wait for the auxiliary policy to converge. Intuitively, the signal for doing this "prospective estimation" comes from quantifying the states at which the auxiliary policy is most quickly converging to the teacher's policy.
>
> 2. **Pretraining the auxiliary policy head**:
> While this is a relatively simple strategy in comparison to our other suggestions, we believe that pretraining the auxiliary policy using offline behavior cloning could make the learning problem more efficient as we will start training of the main policy with the auxiliary policy already well optimized.

---

### Decision · Program_Chairs · 2021-09-27

**Decision:**

Accept (Poster)

**Comment:**

There was a strong consensus among reviewers that this paper should be accepted. The paper proposes a method to address the imitation gap - the setting where the expert has access to privileged information making imitation impossible for the student. The paper shows both the efficacy of the method and the importance of the imitation gap for performance. They only point of improvement would be to look at more diverse environment. Currently the paper is focused on visual navigation.